# Snow model comparison to simulate snow depth evolution and sublimation at point scale in the semi-arid Andes of Chile

Annelies Voordendag[1*], Marion Réveillet[2**], Shelley MacDonell[2], and Stef Lhermitte[1]

[1]Department of Geoscience and Remote Sensing, Delft University of Technology, Delft, The Netherlands
[2]Centro de Estudios Avanzados en Zonas Áridas (CEAZA), ULS-Campus Andrés Bello, Raúl Britan 1305, La Serena, Chile
[*]now at: Department of Atmospheric and Cryospheric Sciences (ACINN), University of Innsbruck, Innsbruck, Austria
[**]now at: Univ. Grenoble Alpes, Université de Toulouse, Météo-France, CNRS, CNRM, Centre d'Etudes de la Neige, 38100 Grenoble, France

**Correspondence:** Shelley MacDonell (shelley.macdonell@ceaza.cl) and Stef Lhermitte (s.lhermitte@tudelft.nl)

**Abstract.** Physically-based snow models provide valuable information on snow cover evolution and are therefore key to provide water availability projections. Yet, uncertainties related to snow modelling remain large as a result of differences in the representation of snow physics and meteorological forcing. While many studies focus on evaluating these uncertainties, no snow model comparison has been done in environments where sublimation is the main ablation process. This study evaluates a case study in the semi-arid Andes of Chile and aims to compare two snow models with different complexities, SNOWPACK and SnowModel, at a local point, over one snow season and to evaluate their sensitivity relative to parameterization and forcing. For that purpose, the two models are forced with i) the most ideal set of input parameters, ii) an ensemble of different physical parameterizations and iii) an ensemble of biased forcing. Results indicate large uncertainties depending on forcing, the snow roughness length $z_0$, albedo parameterization and fresh snow density parameterization. The uncertainty caused by the forcing is direcly related to the bias chosen. Even though the models show significant differences in their physical complexity, the snow model choice is of least importance, as the sensitivity of both models to the forcing data was in the same order of magnitude and highly influenced by the precipitation uncertainties. The sublimation ratio ranges are in agreement for the two models: 36.4 to 80.7% for SnowModel and 36.3 to 86.0% for SNOWPACK, and are related to the albedo parameterization and snow roughness length choice for the two models.

## 1 Introduction

Snow models provide valuable information on snow cover evolution and are therefore key to quantify runoff and provide accurate water availability projections. Several models, with different complexities in the representation of different snow processes, from empirical to physically-based approaches, have been developed to simulate snow depth changes. Empirical approaches, such as degree-day models (e.g. Braithwaite and Olesen, 1989; Hock, 2003) are based on a simple statistical relationship to positive air temperatures to simulate snow melt. Comparatively, physically-based approaches consider all energy flux exchanges at the snow surface by solving the surface energy balance equation (Oke, 2002). The use of the energy balance

equation, coupled with snow models, enables a more complete understanding of snow physical processes and are essential for understanding the interaction between snow cover evolution and climate change.

Physically-based snow models have different complexities in their physical representations, from a single layer approach (e.g. Strasser and Marke, 2010), to more sophisticated multi-layer detailed models representing the evolution of snow microstructure and the layering of snow physical properties (e.g. Bartelt and Lehning, 2002; Vionnet et al., 2012), leading to a wide variety of snow models with a wide variety of parameterizations. In a snow model intercomparison study, Etchevers et al. (2004) highlighted the importance of parameterization choice, especially regarding the net longwave and albedo characterisation. After comparing 33 snow models, Rutter et al. (2009) concluded that no universal 'best' model exists and model performance mainly depends on the study site. Furthermore, the Earth System Model - Snow Model Intercomparison Project (ESM-SnowMIP) compared several snow models to improve the models in the context of local- and global scale modelling (Krinner et al., 2018) and indicated scientific and human errors in snow model intercomparisons (Menard et al., 2021), but the study sites did not include semi-arid regions.

In addition to the development of new models, many studies have focused on model improvements offering different parameterizations in a single model (e.g. Douville et al., 1995; Dutra et al., 2010; Essery, 2015). In such frameworks, many parameters need to be calibrated and are often difficult to be set according to local measurements, such as the albedo and aerodynamic roughness length (Brock et al., 2000, 2006). To address this issue, and to consider and quantify parameter uncertainty propagation in simulated snow depth changes, recent studies have started to use ensemble approaches. Here models are evaluated based on different likely combinations of values of variables such as snow albedo, snow compaction, fresh snow density and liquid water transport (e.g. Essery et al., 2013; Lafaysse et al., 2017; Günther et al., 2019).

In addition, forcing data uncertainty has a significant influence on the simulated snow depth changes (e.g. Magnusson et al., 2015; Raleigh et al., 2015; Günther et al., 2019) and needs to be considered in model evaluations. While point scale simulations forced by direct observations generally reduce forcing uncertainties, measurement errors can be considerable due to the complexity of both measuring certain parameters as well as maintaining measurement sites (e.g. for precipitation (MacDonald and Pomeroy, 2007; Smith, 2007; Wolff et al., 2015), sensor inclination (Weiser et al., 2016) or sensor failure). Methods such as stochastic perturbation with random noise (e.g. Charrois et al., 2016) or following a uniform or normally distributed bias with different magnitudes (e.g. Raleigh et al., 2015) can be used to build an ensemble of meteorological forcing and explicitly simulate the impact of forcing uncertainty on the simulated snow depth (e.g. Charrois et al., 2016; Zolles et al., 2019; Günther et al., 2019).

Despite past efforts to improve snow models and quantify uncertainty propagation, the uncertainties regarding snow physics representation and meteorological forcing remains (e.g. Essery et al., 2013; Raleigh et al., 2015; Günther et al., 2019); in particular in regions where sublimation is the main ablation process, due to the lack of snow modelling studies in semi-arid regions (Gascoin et al., 2013; Réveillet et al., 2020; MacDonell et al., 2013a; Mengual Henríquez, 2017).

This study aims to evaluate two physical snow models with different complexities, considering parameterization and forcing uncertainties. We simulate snow depth changes in the semi-arid Andes of Chile using data from an automated weather station. In this region, snow model uncertainty is a key concern as snow melt is an essential water resource for the population (Favier

et al., 2009). Despite the importance of snow as water resource, quantifying and understanding the snow cover evolution remains limited and challenging due to i) high sublimation rates related to high levels of incoming solar radiation, cold air temperatures, arid atmosphere, and high wind speeds (e.g. MacDonell et al., 2013a; Réveillet et al., 2020), and ii) shallow snow depths due to very low precipitation amounts (Scaff et al., 2017; Réveillet et al., 2020; Ayala et al., 2017). In previous studies, Gascoin et al. (2013) assessed the effect of wind transport on snow cover in the semi-arid Andes using numerical simulations with SnowModel (Liston and Elder, 2006b), and highlighted the significant importance of blowing snow sublimation. They also evidenced the difficulty of the model to capture the small-scale snow depth spatial variability, partly related to the lack of reliable input data such as precipitation. Réveillet et al. (2020) indicated that ablation is dominated by sublimation in the semi-arid Andes and that the sublimation ratio increases with elevation. They also quantified a similar proportion of sublimation ratio for two years with contrasting climatic conditions (i.e. dry versus wet), but pointed out the significant uncertainties related to the forcing. The study performed by Mengual Henríquez (2017) assessed the snow types in different Chilean regions with SNOWPACK (Bartelt and Lehning, 2002; Lehning et al., 2002a, b) and mainly found that SNOWPACK is a powerful snow model, but an improvement of the forcing data is needed to improve simulations. Despite these previous studies, an accurate assessment of different snow models' sensitivity to parameterization choice or input forcing is currently missing, although it is expected to have a large impact.

In this work, the sensitivity of SnowModel and SNOWPACK, the common snow models previously used in this region, is assessed based on parameterization choices and forcing uncertainty. First, the models are calibrated similarly to allow later comparisons and a most ideal setup for both models is designed to acquire a precipitation dataset that corrects for the underestimation of precipitation. Second, both models are run with different combinations of parameterizations to assess the uncertainty of parameterizations. Subsequently, forcing uncertainty propagation in the snow model is considered by running the models with 1000 sets of perturbed forcing. The combination of sensitivity analysis to model parameterizations and meteorological forcing allows to evaluate and compare the two models.

## 2   Study area and data

We assess the sensitivity of both models using data from an AWS over the snow season of 2017. First this study area and the meteorological observations are described, followed by the data preprocessing procedure.

### 2.1   Study area

The study area is located in the La Laguna catchment, in the Chilean Coquimbo region, close to the Argentinian border (Fig. 1a). To assess the sensitivity of the models to the representation of snow physics and meteorological forcing, we use data from the Tapado Automatic Weather Station (AWS), a permanent meteorological tower since 2009 located close to the terminus of the Tapado Glacier at 30°S, 69°W, 4306 m a.s.l. (Fig. 1c). The site shows a complex topographic setting with average (maximum) wind speeds of 4.2 m s$^{-1}$ (>15 m s$^{-1}$) in 2017 and little precipitation (<200 mm a$^{-1}$) that falls as snow during fewer than 10 events per year. Precipitation events mainly occur during the winter season (>90%) (Rabatel et al., 2011; Réveillet et al.,

2020). Therefore the area surrounding the AWS is only covered with snow in austral winter. At this elevation, vegetation is

90 extremly sparse.

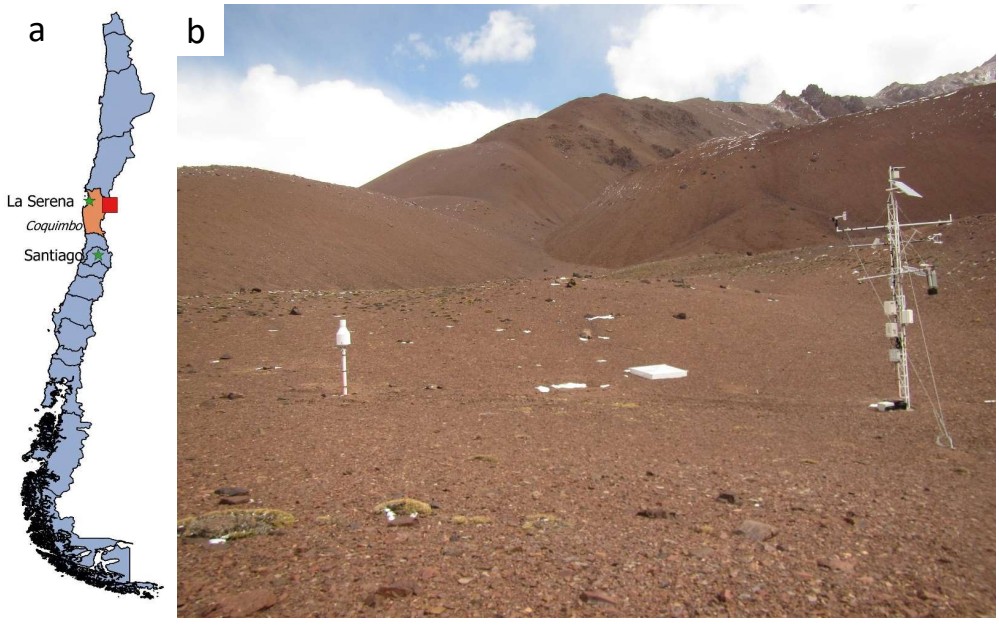

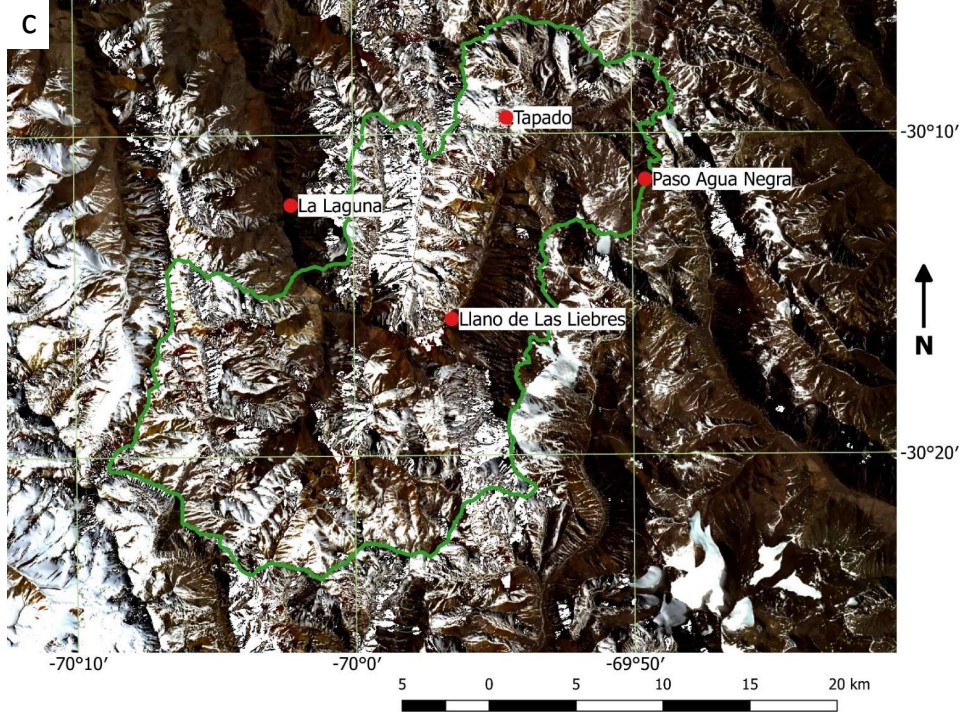

**Figure 1.** a) Map of Chile with the Coquimbo Region (orange) and the study area location (red box). b) Tapado AWS at the 26th of April 2018 showing the Geonor precipitation gauge (left) which is 10 m from the central mast of the AWS (right). c) Map of the borders of the La Laguna catchment (green), with the AWS locations (red points). Landsat 8 images of 29 August 2017 are used as background and maps and photo are made by A. Voordendag.

## 2.2 Meteorological observations

The meteorological forcing data consisted of hourly mean values of air temperature (TA), relative humidity (RH), incoming shortwave radiation ($S_\downarrow$), incoming longwave radiation ($L_\downarrow$), wind speed (WS), wind direction (WD) and air pressure (PA) measured by the AWS (Fig. 1b, 2, Table 1). Precipitation forcing (P) consisted of hourly data by a Geonor rain gauge (Fig. 1b). This gauge is an unshielded, unheated weighing bucket precipitation gauge filled with anti-freeze liquid and oil to prevent freezing and evaporation respectively. During the snow season, defined as the period with snow on the ground (i.e. between 10 May and 6 November 2017), the station recorded meteorological observations continuously except for the TA and RH, for which gaps have been filled using three nearby AWSs (Fig. 1c, Sect. 2.3).

**Table 1.** Available observations, sensor height from the ground and the manufacturers and type of the corresponding sensor at Tapado AWS.

| Measurement | Unit | Height (m) | Brand/type | Uncertainty given by manufacturer |
|---|---|---|---|---|
| Accumulated precipitation (P) | mm | 1.5 | Geonor/T-200B 1000mm | 0.1% Full Scale |
| Air pressure (Pa) | hPa | 3.5 | Vaisala/PTB110 | $\pm$1.0hPa |
| Air temperature (TA) | °C | 3.5 | Vaisala/HMP45C | $\pm 0.3$°C at 0°C |
| Incoming LW radiation ($L_\downarrow$) | W m$^{-2}$ | 3.5 | Kipp and Zonen/CNR4 | 10% (95% confidence level) |
| Incoming SW radiation ($S_\downarrow$) | W m$^{-2}$ | 3.5 | Kipp and Zonen/CNR4 | 5% (95% confidence level) |
| Outgoing LW radiation ($L_\uparrow$) | W m$^{-2}$ | 3.5 | Kipp and Zonen/CNR4 | 10% (95% confidence level) |
| Reflected SW radiation ($S_\uparrow$) | W m$^{-2}$ | 3.5 | Kipp and Zonen/CNR4 | 5% (95% confidence level) |
| Relative Humidity (RH) | % | 3.5 | Vaisala/HMP45C | $\pm$2% RH (0 to 90% RH) $\pm$3% RH (90% to 100% RH) and $\pm$0.05% RH/°C |
| Wind speed (WS) | m s$^{-1}$ | 5.4 | RM Young/5103 | $\pm$0.3 m/s |
| Wind direction (WD) | ° | 5.4 | RM Young/5103 | $\pm 3$° |
| Snow depth (SD) | m | 3.5 | Campbell/SR50A | $\pm$1 cm |
| Water equivalent (SWE, thallium, Tl) | mm | 3.5 | Campbell/CS725 | $\pm$15 mm from 0 to 300 mm $\pm$15% from 300 to 600 mm |
| Water equivalent (SWE, potassium, K) | mm | 3.5 | Campbell/CS725 | $\pm$15 mm from 0 to 300 mm $\pm$15% from 300 to 600 mm |

Hourly snow depth (SD), reflected shortwave radiation ($S_\uparrow$) and six-hourly means of snow water equivalent (SWE) were also recorded at the station and used for model calibration and evaluation. SWE was measured with a CS725 sensor by Campbell Scientific which passively detects the change in naturally emitted terrestrial gamma radiation from the ground after it passes through snow cover. It provided two independent SWE observations measuring both potassium and thallium gamma rays (Wright, 2011). The uncertainty given by the manufacturer is $\pm$15 mm from 0 to 300 mm and $\pm$15% from 300 to 600 mm, but differences of up to 82 mm w.e. between potassium and thallium gamma ray measurements at $\sim$300 mm w.e. were measured.

The manufacturer suggests that the output with the higher count is generally the most reliable, which were the potassium gamma rays measurements (Randall, 2018, personal communication). We display both data sets and estimate an uncertainty of ±25 mm for this data set.

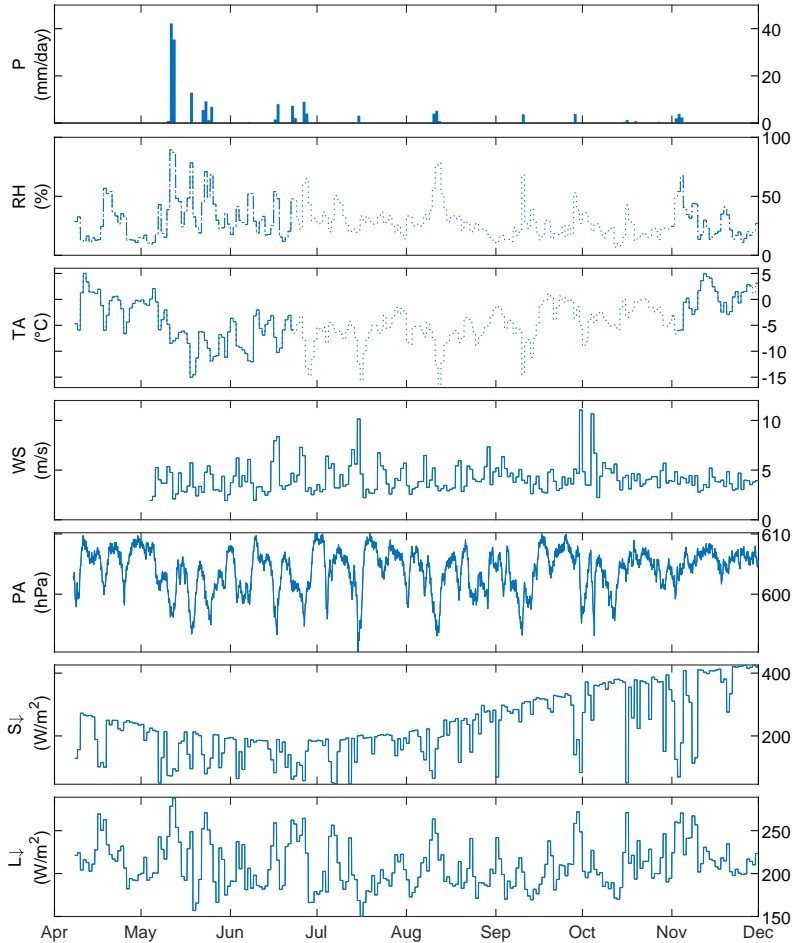

**Figure 2.** Meteorological observations at Tapado with with daily precipitation (P), average daily relative humidity (RH), air temperature (TA), wind speed (WS), air pressure (PA), incoming shortwave radiation ($S_\downarrow$) and incoming longwave radiation ($L_\downarrow$) from April to December 2017. Dotted lines indicate the TA and RH interpolations.

## 2.3 Preprocessing of forcing data

The period between 5 May and 30 November 2017 has been covered to model the snow evolution in the austral winter, as this
was a season where SWE data was available to validate the models. Since continuous data are required for both snow models,
preprocessing was necessary to fill the gaps in the TA and RH data sets (23 June 11:00 to 31 October 10:00 due to sensor
failure) and to correct the wind-induced undercatch in the precipitation data. Therefore, TA and RH data were interpolated
based on lapse rates from nearby AWSs (Agua Negra (4774 m a.s.l.), Llano de las Liebres (3565 m a.s.l.) and La Laguna (3209
m a.s.l.; Fig. 1c).

For TA, a daily temperature lapse rate (Blandford et al., 2008) was calculated using TA measured at La Laguna and Paso
Agua Negra AWSs (1565 m elevation difference) between 2014 and 2017. We fitted a sinusoidal trend over these lapse rates
for the four-year period and found daily lapse rates with a maximum of -6.9°C km$^{-1}$ in winter and a minimum of -8.0°C km$^{-1}$
in summer. These daily lapse rates were subsequently applied to TA observations of Llano de las Liebres AWS which is the
only AWS that covers the entire period of missing data in 2017. For RH a similar approach was applied using the lapse rate
of the daily dew point temperature between the Paso Agua Negra and La Laguna AWSs and applying it to data measured at
the Llano de las Liebres AWS. Dew point temperature was converted to RH following Liston and Elder (2006a). Evaluation
of this lapse rate interpolation, based on 1638 overlapping observations at Tapado, shows an uncertainty (i.e. RMSE) of 2.8°C
and 9.97% for TA and RH respectively.

Since the precipitation observations were directly influenced by wind, an undercatch in the precipitation gauge is likely (e.g.
MacDonald and Pomeroy, 2007; Smith, 2007; Wolff et al., 2015). As there are different possibilities to correct for this, the
assimilation and correction of precipitation data is explained in Sect. 3.2.2.

## 3 Methods

### 3.1 Model descriptions

#### 3.1.1 SNOWPACK

SNOWPACK was developed by the Swiss Federal Institute (SLF) for Snow and Avalanche Research (Bartelt and Lehning,
2002; Lehning et al., 2002a, b). It is a one-dimensional model, but can be implemented in the spatially distributed, three-
dimensional snow cover and earth surface model *Alpine3D* (Lehning et al., 2006). SNOWPACK includes the *MeteoIO* pre-
processing library for meteorological data (Bavay and Egger, 2014) which was not used, as we implemented a homogeneous
preprocessing approach for both models (see Sect. 2.3). SNOWPACK is a physically-based model which has the ability to
simulate snow physical properties (e.g. snowpack temperature, layer thickness, snow microstructure and density) and snow
processes (e.g. refreezing, sublimation, melt, evaporation) for multiple layers, which are merged if layers become too thin.
Sublimation and evaporation are calculated for the top element of the snowpack and melt is simulated using a water transport
bucket scheme. In this bucket scheme, all the liquid water exceeding a threshold water content is transported downward in

the snowpack or soil (Wever et al., 2014). An extensive description of the model can be found in Bartelt and Lehning (2002);
Lehning et al. (2002a, b).

### 3.1.2 SnowModel

SnowModel is a spatially distributed snowpack evolution modelling system composed of four submodels *MicroMet*, *EnBal*,
*SnowPack* and *SnowTran3D* (Liston and Elder, 2006b). MicroMet is a preprocessing library for meteorological data interpola-
tion, which was not used in this study as we focused on one location only while we implemented a homogeneous preprocessing
approach for both models (see Sect. 2.3). EnBal calculates standard surface energy balance exchanges (Liston and Hall, 1995).
SnowModel's SnowPack subroutine is a single or multi-layer (max. six layers) snowpack evolution and runoff model that de-
scribes snowpack changes in response to precipitation and melt fluxes defined by MicroMet and EnBal (Liston and Hall, 1995;
Liston and Elder, 2006b). In SnowModel, the melted snow is redistributed over the remaining snow depth up to a maximum
density threshold of 550 kg m$^{-3}$. Any additional melt water is added to the runoff. In this study the model was run with the
maximum of snow layers (i.e. six layers) to be comparable with the multiple amount of layers in SNOWPACK. Finally, the
three-dimensional model SnowTran3D (Liston and Sturm, 1998), which simulates snow erosion and deposition, is not activated
in this study; this choice is discussed in Sect. 5.2.

## 3.2 Model setup and sensitivity analysis

To assess the sensitivity of both models to parameterization choice and input uncertainty, we applied a four step approach.
First, we set up both models similarly to allow later comparisons (Sect. 3.2.1). Second, we designed the most ideal setup
for both models to acquire a precipitation dataset that corrects for the underestimation of precipitation. Third, we varied the
parameterization settings of each model to determine the effect of parameterization choice (Sect. 3.2.3). Last, we implemented
forcing biases (Sect. 3.2.4) to evaluate the sensitivity of the models to the meteorological forcing uncertainties. The combination
of sensitivity analysis to model parameterizations and meteorological forcing allowed to evaluate and compare the two models
(Sect. 3.3).

### 3.2.1 Parameter values used in both models

Initially, both models were set up using similar parameters to facilitate intercomparison. These parameters were derived from
observations or previous studies (Table 2). For example, the soil albedo was set to 0.15, as this is the observed surface albedo
when there is no snow cover. The observed daily albedo is defined as the daily sum of the average hourly $S_\uparrow$ divided by the
daily sum of average hourly $S_\downarrow$ (Fig. 4e,f). In the absence of roughness length measurements, the roughness length of the bare
soil is set to 0.020 m, corresponding to the default roughness length of pebbles and rocks in SnowModel. As surface ground
temperature measurements are not available, we set it to -1°C in both models. -1°C is the default value in SnowModel and
ensures that the fresh snow does not immediately melt.

**Table 2.** SNOWPACK and SnowModel parameter characteristics. The possible snow albedo parameterizations and fresh snow density models are described in Sect. S1.

| | SNOWPACK | SnowModel |
|---|---|---|
| Soil albedo | 0.15 (calibrated) | 0.15 (calibrated) |
| Max/min snow albedo | None | 0.6/0.9 (calibrated) |
| Atmospheric stability correction model | Richardson number | Richardson number (default) |
| Roughness length (soil) | 0.02 m | 0.02 m (default) |
| Roughness length (snow) | 0.001 m and 0.01 m | 0.001 m and 0.01 m |
| Surface ground temperature | -1°C | -1°C (default) |
| Thermal conductivity | Default | Multilayer subroutine |
| Wind erosion/snow transport by wind | Off | Off |
| Maximum number of snow layers | Unlimited | 6 layers |
| Fresh snow density parameterizations | 5 options | 1 default option and 2 from SNOWPACK |
| Albedo parameterizations | 6 options, 4 used | 2 options |
| Simulated ablation processes | Sublimation, runoff, evaporation | Sublimation, runoff |
| Water transport in snowpack | Bucket scheme (default) | Default |

### 3.2.2 Idealised setup

Preliminary results showed simulated SWE and SD to be more than two times lower than the observed SWE. This is caused by an underestimation of the precipitation measurements, as the AWS is placed in a concave area that collects more snow than the Geonor precipitation gauge. This is in correspondence with research by Grünewald and Lehning (2014) on the spatial variability of SD measurements. Therefore, to correct the precipitation used as input for the models, an idealised setup is designed, making use of all the data that the models allow as forcing. Two approaches are designed to adjust the precipitation data set. First, it is

chosen to assimilate a precipitation data set, which both models perform in different ways. SNOWPACK assimilates the data if SD is given as input. Reflected SW radiation is also given as input, to prevent inaccurate parameterizations of the albedo. The precipitation data set is assimilated with the five possible fresh snow density parameterizations in SNOWPACK. SnowModel allows the possiblity to assimilate the precipitation when SWE is given, but is not able to cope with reflected SW radiation as input. Therefore, six ensembles are made out of two albedo and three fresh snow density parameterizations to find an

assimilated precipitation data set. Second, the precipitation is reconstructed from the SWE observations, which was computed using the cumulative positive SWE (potassium) changes during precipitation events (detected by the Geonor T-200B). The positive SWE changes beyond precipitation events are not accounted for, as they might originate from deposition caused by snow drift and its inclusion would have resulted in an overestimation in this data set. Hereinafter this precipitation dataset is called PSWE.

Atmospheric stability and snow roughness length ($z_0$) are key parameters in semi-arid regions where sublimation is an important process. As SnowModel only allows atmospheric stability corrections based on the Richardson number, we opted for this method and similar roughness lengths in both models to assure intercomparability. The $z_0$ was set to both 0.001 m and 0.01 m. 0.001 m is based on an earlier sensitivity study (Réveillet et al., 2020) and unpublished eddy covariance measurements (MacDonell et al., 2013a); 0.01 m is based on literature (e.g Brock et al., 2006; Cuffey and Paterson, 2010). The first and second approach of the idealised setup are both tested with $z_0$ of 0.001 m and 0.01 mm, thus the idealised setup in total consists of four cases of each five (SNOWPACK) or six (SnowModel) simulations.

This idealised case corresponds therefore to simulations using the best possible combination of input data. As such observations are not always available or used to evaluate models, the idealised simulations are not used for the sensitivity study and model comparisons, which are based on optimal simulations (i.e. without assimilating precipitation and albedo, see Sect. 3.2.3). The simulated SWE and SD are compared to the observed SWE and SD and the assimilated precipitation data sets are shown in Fig. 3. Due to complexities with the assimilated precipitation data and the need for SWE as validation data, the precipitation data set ($P_{cor}$) that is used in the further study is based on a wind correction by Wolff et al. (2015) (See Sect. 4.1).

### 3.2.3 Sensitivity analysis of variable parameterizations

To assess the impact of the parameterizations on the snowpack simulation, an ensemble approach based on different combinations of albedo and snow density parameterizations and $z_0$ was used (e.g. Essery et al., 2013; Lafaysse et al., 2017, and Sect. 3.2.2). The choice to limit the sensitivity tests to these three parameters is discussed in Sect. 5.2.

For SNOWPACK, 40 runs were performed over the 2017 season based on four different albedo, five fresh snow density parameterizations and two different $z_0$ values. Each of the albedo parameterizations is based on empirical relations derived from continuous observations at Weissfluhjoch (Lehning et al., 2002a) or on grain size (Schmucki et al., 2014), while the fresh snow parameterizations are empirical formulas depending on the TA, RH, WS and surface temperature. More details are found in the Sect. S1 (Supplementary Material (SM)) and the mentioned references.

For SnowModel, an ensemble of 12 simulations was run, considering two albedo, three snow density parameterizations and two different $z_0$. The albedo parameterizations range between 0.6 and 0.9 depending i) on TA solely (more details in Liston and Hall (1995), Liston and Elder (2006b) and in Sect. S1 (SM)) or ii) on TA and time (Strack et al., 2004, and Sect. S1). SnowModel's default fresh snow density parameterization depends on the wet bulb temperature, but we included two fresh snow density parameterizations from SNOWPACK depending on TA, RH, WS and surface temperature to test the model more extensively. In these additional parameterizations, we preserved the SnowModel defaults for minimum (50 kg m$^{-3}$) and maximum fresh snow density (158.5 kg m$^{-3}$).

Each of the ensemble simulations was forced by the observations (TA, RH, PA, WS, WD, $S_\downarrow$, $L_\downarrow$) as described in Sect. 2.3 and the $P_{cor}$ acquired after the idealised setup (Sect. 3.2.2). The simulations are evaluated by comparing the model output of SD, SWE and albedo with the corresponding observations. Based on this evaluation the simulation with the lowest $RMSE$ and highest $R^2$ between the observed and modelled albedo is chosen as the reference for the forcing sensitivity analysis discussed in Sect. 3.2.4.

**Table 3.** Forcing data for the snow models with the corresponding uncertainty $\sigma$ used in the ensemble simulation. The ranges of PA, TA, $S_\downarrow$, $L_\downarrow$, RH and WS are ranges as used by Raleigh et al. (2015). The WD range is according to the uncertainty given by the manufacturer and the $P_{cor}$ range is described in Sect. 3.2.4.

| Forcing | Distribution | Range | Unit |
|---|---|---|---|
| Accumulated precipitation ($P_{cor}$) | Uniform | [-100,+100] | mm a$^{-1}$ |
| Air pressure (PA) | Normal | [-100,+100] | Pa |
| Air temperature (TA) | Normal | [-3.0,+3.0] | °C |
| Incoming longwave radiation ($L_\downarrow$) | Normal | [-25,+25] | W m$^{-2}$ |
| Incoming shortwave radiation ($S_\downarrow$) | Normal | [-100,+100] | W m$^{-2}$ |
| Relative humidity (RH) | Normal | [-0.25,+0.25] | % |
| Wind speed (WS) | Normal | [-3,+3] | m s$^{-1}$ |
| Wind direction (WD) | Normal | [-3,+3] | ° |

### 3.2.4 Forcing uncertainty estimation

To assess the model sensitivity to meteorological measurement uncertainties, a bias has been applied to the meteorological forcing presented in Sect. 2.3 to generate an ensemble of 1000 forcing files. Raleigh et al. (2015) have shown that the model outputs are more sensitive to forcing biases than random errors. Therefore, all input variables except $P_{cor}$ were modified by adding hourly biases with a normal distribution $N(\mu = 0, \sigma^2)$ with $\sigma$ the uncertainty range taken from Raleigh et al. (2015) and reported in Table 3. The biases has been kept within their corresponding range (Table 3) by assuming that the 99.7% of the bias, 225 thus $3\sigma$, is within this range. This positive component of the range is divided by three and multiplied with a normally distributed random number and added to the observed forcing. We have chosen 1000 runs as a compromise between computational effort and a reliable confidence interval.

The distribution of the precipitation uncertainty is chosen to be uniform, as the observed precipitation was low (i.e. 180.7 mm w.e. at the end of the season) and the differences between the assimilated precipitation (SnowModel) and PSWE cover 230 approximately 200 mm w.e. (Sect. 4.1).

Subsequently, based on the perturbed input data, 2000 snow model simulations are performed: 1000 with meteorological biases and 1000 with combined meteorological/precipitation biases. This setup was chosen to enable the differentiation between meteorological and precipitation uncertainties, which would be difficult in a combined approach where precipitation uncertainty would dominate.

## 3.3 Model evaluation

Model evaluation consists of comparing the model output of SD, SWE and albedo with the corresponding observations. For the idealised case this consists of evaluating the $RMSE$ and $R^2$ between the modelled and the observed SD to acquire precipitation data that approaches the correct mass balance. For the parameterization uncertainty, this consists of evaluating the $RMSE$ and

$R^2$ between the modelled and the observed albedo, to select the best reference for each model (i.e. 40 for SNOWPACK and 12 for SnowModel). In this case, it is chosen to only compare between modelled and observed albedo, as this ensures the best possible net shortwave radiation term in the energy balance equation. The forcing uncertainty is evaluated by comparing the differences of end of snow season. Last, the differences in ablation processes of the parameterizations are shown.

## 4 Results

### 4.1 Idealised simulations

The assimilated precipitation datasets markedly differ between SNOWPACK and SnowModel (blue lines, Fig. 3e,f). For clarity Fig. 3 shows only the results of the idealised simulations for the $z_0$ value of 1 cm; the results for $z_0$ of 1 cm and 1 mm are displayed in Sect. S2. For SNOWPACK, eight out of ten runs with $z_0$=1 cm crashed, thus only two simulations are shown. The reason for these crashes has not been further investigated.

Assimilation of SD in SNOWPACK results in SWE that approximates the PSWE (Fig. 3), leading to assimilated precipitation amounts of 2.55 to 3.02 times the observed precipitation and a good correspondence with the observed SD (i.e. $RMSE$ between 9.2 and 11.5 cm and $R^2$ between 0.90 and 0.93 calculated with the observed and simulated SD, Fig. 3a).

Assimilation of SWE in SnowModel only adjusts the precipitation between 22 and 27 June and between 7 and 12 August. The amount of precipitation is not adjusted at the beginning of the season and thus, the assimilated data by SnowModel still leads to an underestimation of the SD and SWE (Fig. 3b,d). The missing adjustment of the SWE is probably caused by SnowModel taking a maximum of 99 SWE observations and the observations do not exactly align with the precipitation events, which leads to correction factors of one (i.e. no change) to the precipitation data. The assimilated precipitation is approx. 1.6 times larger than the observed precipitation and the agreement between modelled and observed SD is better for SNOWPACK than for SnowModel (i.e. $RMSE$ between 7.1 and 17.1 cm and $R^2$ between 0.19 and 0.90 calculated with the observed and simulated SD, Fig. 3b).

Both models overestimate the SWE between mid-July and September when PSWE was given as input (red lines in Fig. 3c,d). This is likely caused by an overestimation of the PSWE at the end of June. Only small amounts of precipitation are observed at the precipitation gauge, but the observed SWE distinctly increases probably due to snow drift, as strong winds were also observed. A similar thing occurs at the end of September. The models markedly increase the amount of precipitation (between 97 and 137 mm w.e.) in the assimilation runs, but only small amounts of precipitation and strong winds were observed. This is related to a strong melt in September, not simulated by the model, along with models trying to get the SWE and SD to the observational levels. The strong melt in September is caused by a sudden decrease of the albedo (observations in Fig. 4e-f), as it is likely that the snow got covered with dust after some days with strong wind at the end of September, but the simulations overestimate the melt caused by this albedo decrease. Likewise, high wind speeds and a strong decrease of SWE and SD are observed mid-July, which is likely due to snow erosion that is not considered in our simulations. The overestimation of and the need for SWE as validation data are indications that the PSWE is not a valid precipitation dataset for our simulations, but

it is also unfeasible to select one of the assimilated precipitation sets by SNOWPACK as the amount of precipitation markedly increases at the end of September and we want to use SD as validation data.

Therefore, three different precipitation corrections depending on WS (Smith, 2007; MacDonald and Pomeroy, 2007) or on TA and WS (Wolff et al., 2015) were applied to the observed precipitation (See Sect. S3). Eq. (12) from Wolff et al. (2015)
with WS corrected to gauge height using a logarithmic wind profile (e.g. Lehning et al., 2002a) and a $z_0$ of 0.01 m is used as precipitation data ($P_{cor}$) in the further study, as this correction approaches the PSWE and shows an increase in precipitation of 2.35 times the observed precipitation at the end of the season.

## 4.2 Sensitivity analysis of parameterizations

Evaluation of simulated SD and SWE based on various parameterizations shows that both models are in good agreement
with observations (Fig. 4), although they overestimate SWE at the beginning of the season (May/June). The correction of the precipitation with the equation from Wolff et al. (2015) overestimates the precipitation in this period, and also leads to an overestimation in the simulations.

For SNOWPACK, the spread of the simulated SD from the 40 different parameterizations (20 simulations for $z_0 = 1$ mm and 20 for $z_0 = 1$ cm) is the largest at the end of the snow season (i.e. October) (Fig. 4a). The date of snow-free surface is
285 the earliest at 8 October and exceeds the simulated period (i.e. after 30 November), depending on parameterization choice, and covers the observed date of snow removal (i.e. 16 October). The different SNOWPACK parameterizations (equations in Sect. S1) show a mean SD difference of 32 cm (which corresponds to 28.9% of the total SD) between the minimum and maximum simulated SD (Fig. 4a), with a maximum of 127 cm observed at 27 June. For the SWE, this corresponds to a mean difference of 98.3 mm w.e. (i.e 28.2% of the total SWE) (Fig. 4c). The large modelled SD spread in May and June can be
explained by the different density parameterization choices as it is not apparent in the SWE simulations (Fig. 4a,c). The rapid decrease (3-8 cm d$^{-1}$) of snow depth until July, caused by compaction of the snowpack, is simulated by the majority of fresh snow density parameterizations, while only one fresh snow density parameterization models a more moderate compaction (i.e. the bold red line has a moderate slope, compared to the light red lines in Fig. 4a, until July). From July onward, the measured snow depth decreases 10 centimetres per 25 days, which is only simulated by the fresh density parameterization that simulated
moderate compaction before July. Snow density measurements were unavailable in 2017 and the observed snow density in Fig. 4g is calculated with $SWE/SD$. The observed snow density is only shown until the end of August, as the calculation led to unrealistic decreasing snow densities after August. This is likely caused by higher readings at the SD sensor than at the SWE sensor, as those sensors were placed on different sides of the meteorological tower or to a bias of the SWE sensor in the ablation season as explained by Smith et al. (2017).
The albedo evaluation (Fig. 4e) and corresponding statistics (Sect. S4) highlight one parameterization choice that outperforms all other parameterizations (i.e. $RMSE$ of 0.09 (-) and $R^2$ of 0.86 calculated with the observed and simulated albedo) in terms of snow compaction after snowfall events, end of snow season, and albedo evolution (Fig. 4e). Therefore this simulation with a $z_0$ of 1 cm is selected as the reference simulation (represented in bold lines in Fig. 4) for the forcing uncertainty simulations.

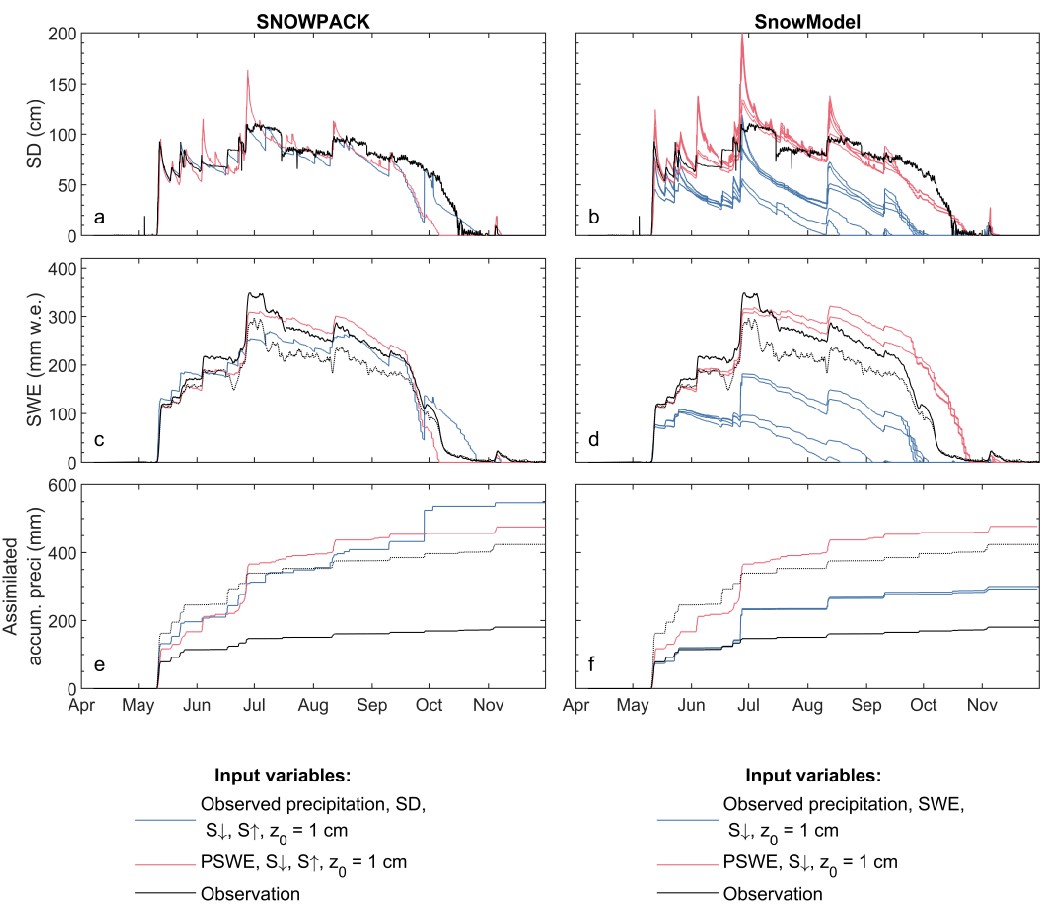

**Figure 3.** a-b) SD, c-d) SWE and e-f) the cumulative assimilated precipitation for the simulations with SNOWPACK (a,c,e) and SnowModel (b,d,f) and observations (black). The different input variables are given in the legenda. The solid (dotted) line in c-d) indicates the more (less) reliable SWE measurement from potassium (thallium) rays (See Sect. 2.3) and the dotted line in e-f) is $P_{cor}$. The models have assimilated the observed precipitation (black) to the output (red/blue) given in e-f). Only one red and one blue line is shown for SNOWPACK as the other eight simulations crashed. The simulations for $z_0 = 1$ mm are found in Sect. S2.

For SnowModel, the largest SD spread of the 12 ensembles (six for every $z_0$, equations in Sect. S1) occurs at the end of the simulated snow season (i.e. August, September and October) with complete snow removal between 21 October and 12 November (i.e. 22 days) (Fig. 4b). The mean SD difference between the parameterizations is 20 cm (i.e. 18% of the total SD), with a maximum of 152 cm at the first snowfall event (12 May), while for SWE the mean difference is 57 mm (i.e. 19.2% of the total SWE) with a maximum of 317 mm w.e. at 12 August (Fig. 4d).

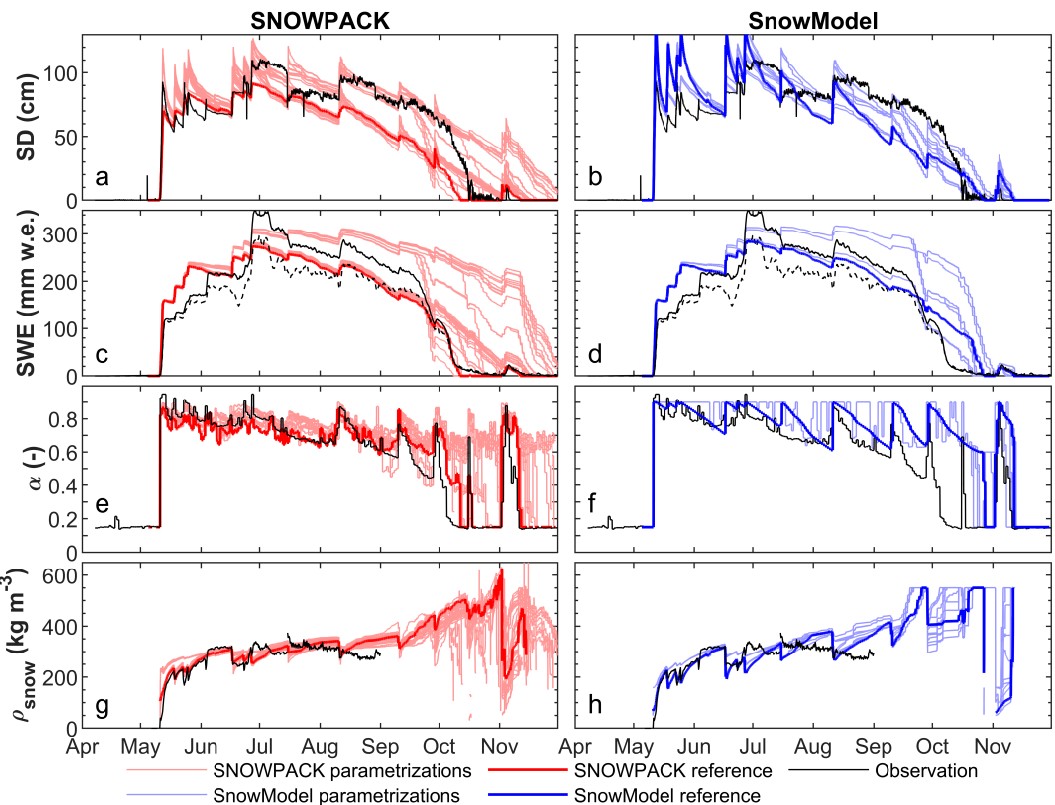

**Figure 4.** a-b) SD, c-d) SWE, e-f) albedo and g-h) snow density simulations (coloured) of the ensemble approaches for SNOWPACK (red) and SnowModel (blue) and observations (black). The bold coloured lines show the reference simulations chosen as the most optimal parameterizaton set according to the measured albedo. The solid (dotted) line in c-d) indicates the more (less) reliable SWE measurement from potassium (thallium) rays.

Quantitative analysis (Sect. S4) shows best performance scores for the time-evolution albedo approach in combination with the reference snow density parameterization and a $z_0$ of 1 cm ($RMSE$ of 0.150 (-) and $R^2$ of 0.600). Therefore, these are used as the reference simulation (bold line in Fig. 4).

Comparison of the SNOWPACK and SnowModel output shows similar SD variations attributed to snow density parameterizations that simulate low density snowfall with notable subsequent compaction (Fig. 3g,h). In reality, this happens at Tapado until June, followed by a different regime with denser fresh snow and less compaction. The biggest difference between the models, however, is the result of the albedo parameterizations. Where SnowModel relies on two albedo models based on TA and albedo ranges, SNOWPACK relies on empirical relations calibrated with measurements in Switzerland and not adapted to the arid Tapado climate. Nevertheless, the albedo of the reference run of SNOWPACK performs well in a semi-arid area. Last, the simulations by SnowModel all approximate the end of snow season within a period of 22 days, whereas the simulations at

the end of season noticeably diverge for SNOWPACK. This is likely caused by TA being above the freezing point at the end of October, resulting in a fast melt simulated for all ensembles by SnowModel.

## 4.3 Sensitivity analysis of forcing data

### 4.3.1 Excluding precipitation uncertainty

The biased forcing excluding precipitation uncertainty shows a similar sensitivity for SNOWPACK and SnowModel (Fig-
ures 5a,c) with mean SD/SWE biases of 52 cm/163 mm w.e. for SNOWPACK and 47 cm/172 mm w.e. for SnowModel. The simulations with SnowModel show more uncertainty in the melting period (e.g. in October), but otherwise, the simulations mainly overlap. The forcing uncertainty results in complete snow removal simulations ranging from 27 August to 28 November (i.e. 93 days) for SNOWPACK and 30 August to 25 November (i.e. 87 days) for SnowModel. The reference simulations of both models are located in the middle of the spread of simulations, which is coherent with the normal distribution of the biases
applied to the forcing. The biggest differences between the models are found in the way SD has been simulated. The reference run and the 1000 simulations with biased forcing show marked settling rates througout the season with SnowModel, whereas the settling is more moderate for SNOWPACK. This depends on the chosen snow density parameterization and is discussed further in Sect. 5.

### 4.3.2 Including precipitation uncertainty

The forcing perturbations including precipitation uncertainty shows that precipitation uncertainty has a large impact on SD and SWE ensemble spread (Fig. 5b,d). Averaged over the season this results in SD/SWE biases of 75 cm/257 mm w.e. and 70 cm/262 mm w.e. for SNOWPACK and SnowModel, respectively. Along with the similar average spread over the entire season observed for both models, the range of the simulated day of snow-free surface is also similar; for SnowModel this date ranges between 20 August and 29 November (i.e. 101 days) and the range is similar but a bit later in the season for SNOWPACK (i.e.
between 29 August and early December). Again, the main differences are found in the settling of the snowpack (See Sect. 5).

## 4.4 Impacts of the model choice and parameterizations on sublimation

Ablation rates (Fig. 6) show that sublimation is the dominant mode of mass loss in both models until September (i.e. cold period), and followed by melt from September to the end of the season (i.e. end of November, and called the melting period). Note that for SNOWPACK, the first day of snow-free surface of the reference run is 11 October and for SnowModel 27 October.
For SNOWPACK, the spread of the averaged sublimation rates corresponding to the ensemble runs from the first day of snow to 20 September have a minimum of 1.41 and a maximum of 2.96 mm w.e. d$^{-1}$ (Fig. 6a). During the cold period, when no melt occurs, the sublimation amounts mainly depend on the $z_0$, with sublimation rates ranging between 1.40 and 3.18 mm w.e. d$^{-1}$, but this is mainly clustered according to the implemented $z_0$. At the end of the season, the total sublimation ranges between 153 and 364 mm w.e. (corresponding to 36.2 to 86.0% of the total ablation, again strongly depending on the $z_0$).
During the melting period, the ensemble runs show a large spread of melt rates ranging between 0.97 to 17.7 mm w.e. d$^{-1}$. The

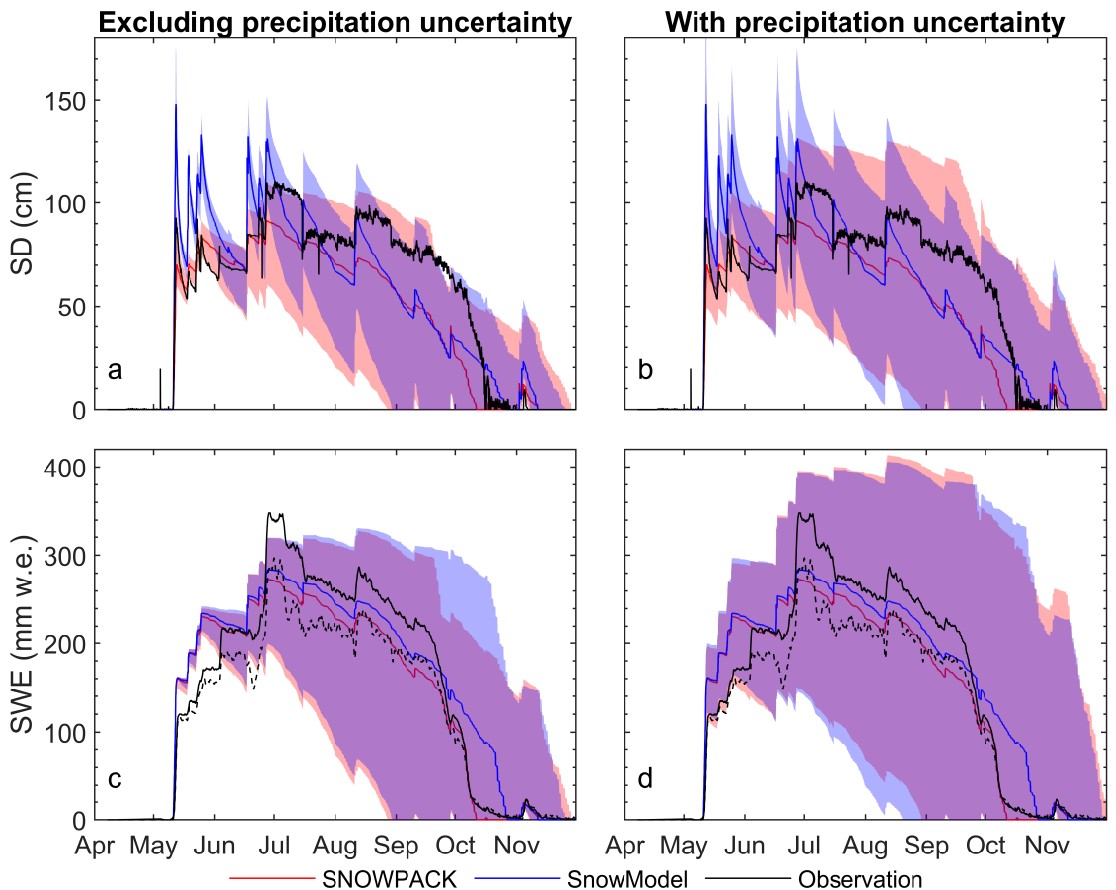

**Figure 5.** Observed (black) and simulated (colour) SD and SWE by SNOWPACK and SnowModel forced by the 1000 ensembles of meteorological data. The reference run (see Sect. 3.2.3) of both models is the bold coloured curve. The shaded area corresponds to the 1000 runs of a)-b) snow depth and SWE c)-d) of SNOWPACK (red) and SnowModel (blue) for the run with biased forcing. a)-c) are excluding precipitation uncertainties and b)-d) are including precipitation uncertainties.The solid (dotted) line in c-d) indicates the more (less) reliable SWE measurement from potassium (thallium) rays.

total amount of runoff is between 28.9 and 236 mm w.e. for SNOWPACK and this model also simulates evaporation, which contributes between 2.5 and 10.2% of total ablation (Fig. 6a).

For SnowModel, sublimation differences between the parameterizations are similar (Fig. 6b) with average sublimation rates from the first day of snow to 20 September ranging between 1.27 to 2.79 mm w.e. $d^{-1}$. At the end of the winter season the sublimation totals range between 154 and 342 mm w.e. (which corresponds to 36.4 to 80.7% of the total ablation). The runoff is between 81.7 and 269 mm w.e.. A closer analysis of Fig. 6b shows that SnowModel's output clusters into four groups, where the grouping is determined by the albedo parameterization and $z_0$ with limited influence of fresh snow density parameterizations.

The two lower clusters are linked to the $z_0$ value of 1 mm. The differences between clusters for different $z_0$ values increase as the difference in albedo between the parameterizations increase at the end of June.

While the ensemble parameterization simulations do not lead to significant differences in the modelled end date of the snow season (i.e. difference of 22 days), the albedo parameterization and $z_0$ value directly impact the proportion of sublimation versus melt to the total ablation (Fig. 6b,d). During the cold period, simulations considering the lowest albedo and $z_0$ of 1 cm (the reference simulation), lead to a higher sublimation rate (Fig. 6b). Indeed, a lower albedo increases the energy absorbed by the snowpack, and as the temperature is below the freezing point, this energy leads to an increase in the sublimation. A higher $z_0$

enhances this process even more as this leads to a more negative latent heat flux. Second, the increase of net shortwave radiation also affects the physical properties of the snowpack resulting in an increase of compaction (Fig. 4b,h). The snow density of the snowpack is therefore higher, which directly affects the thermal conductivity of the upper snow layers (Yen, 1981). Surface temperature variation is directly linked to the latent heat flux and therefore to sublimation, explaining the different sublimation ratios simulated depending on the albedo parameterizations and $z_0$ values.

     In contrast to SnowModel, the albedo parameterization in SNOWPACK does not affect the sublimation but noticeably influences the melting rate (Fig.6c), which can be attributed to the more complex characteristics of this model. SNOWPACK allows refreezing and evaporation of melting snow within the snowpack, which can lead to a longer melt season, whereas calculated evaporation leads to a lower amount of runoff from melt. Also, SNOWPACK considers a more complex representation of snow

physics, such as the grain size and microstructure, which directly impacts the albedo and can help to explain the wide diversity of melt simulations.

## 5   Discussion

### 5.1   Model sensitivity and comparison

Our results show the importance of model parameterizations and model forcing over the snow model choice, despite the limited model options chosen for the ensemble approach, and the large differences in the two model complexities chosen in this study. This conclusion, found here in an arid environment, is in agreement with the studies performed in other alpine areas (Etchevers et al., 2004; Günther et al., 2019).

     The representation of turbulent fluxes is an important variable to consider to simulate sublimation and in snow models this

is commonly based on the bulk method; the Richardson number is often used, together with the Monin–Obukhov similarity theory, to evaluate the atmosphere stability (e.g. Liston and Hall, 1995; Vionnet et al., 2012). Here, only the Richardson number is used as both models offered this option and the uncertainty associated to the turbulent fluxes parameterization is only considered by implementing two different $z_0$ values, while it can have major implications in surface energy balance modelling (e.g. Dadic et al., 2013; Conway and Cullen, 2013; Litt et al., 2017; Réveillet et al., 2020). While the stability function cannot

be compared between the two models chosen in this study, the sensitivity of SNOWPACK to the six possibilities available

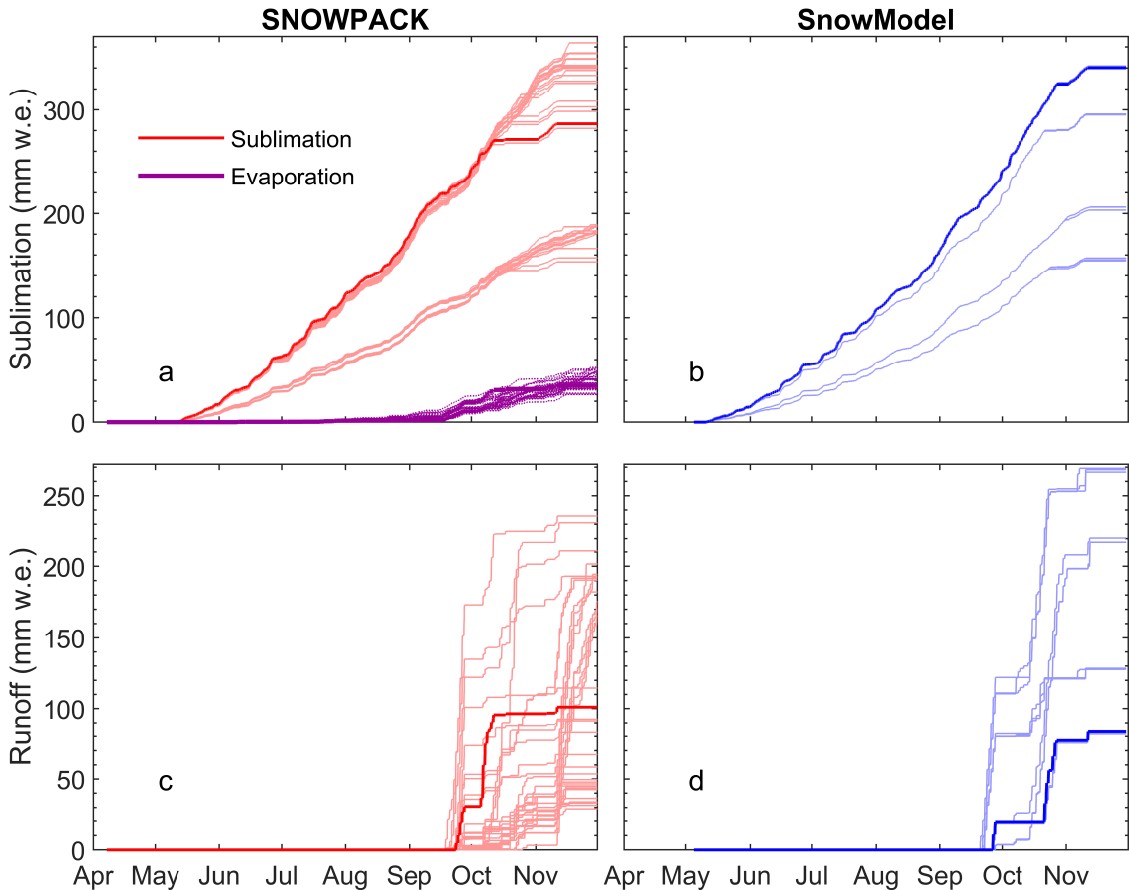

**Figure 6.** Cumulative sublimation (a,b) and runoff from melt (c,d), simulated by SNOWPACK (red) and SnowModel (blue). For SNOW-PACK the cumulative evaporation from melt is shown (purple lines in a). Results for all the ensembles of parameterizations are shown and the bold lines correspond to the reference simulations of each model.

in the current version is low (i.e. max SD bias of a few centimeters, results not shown). In a study over Brewster Glacier in New-Zealand, Conway and Cullen (2013) pointed out the importance of the stability functions to properly simulate the heat fluxes with low wind speed and large temperature gradients and also that the modelled latent heat fluxes were unaffected by the choice of exchange coefficient parameterization. The present study takes place in a dry and windy environment without

a large temperature gradient and this helps to explain the small differences observed related to the different atmospheric stability functions. The turbulent fluxes parameterization is sensitive to the $z_0$ value and observations, such as Eddy Covariance measurements, are essential to accurately parameterize the turbulent fluxes (e.g. Conway and Cullen, 2013; Litt et al., 2017; Réveillet et al., 2018).

Due to the absence of such measurements, the variability of this value over time (e.g. MacDonell et al., 2013b; Pellicciotti

et al., 2005; Nicholson et al., 2016) and due to other values in literature at other locations showing a wide variety (two orders of

magnitude) of the snow roughness length (Gromke et al., 2011; Poggi, 1977; Bintanja and Broeke, 1995; Andreas et al., 2005), it was decided to use two different values for $z_0$ (1 mm and 1 cm). Similar sensitivity ranges for SNOWPACK and SnowModel were found (e.g. Fig. 4), along with similar sublimation rates, but this directly depended on the value for $z_0$. For both models, a $z_0$ of 1 cm led to better simulations (Fig. 3, 4), but as applied more often, this can also be seen as optimizing parameter (e.g. Stigter et al., 2018). In future work, the $z_0$ can be evaluated with eddy covariance measurements.

The biggest differences between the models are found as the snow settles and therefore depends on the snow density parameterization. The challenge in this study was that the snow settling showed two distinct regimes. From May to mid-June, high compaction rates were found, whereas the compaction afterwards was more moderate. SNOWPACK is able to model both moderate and high compaction, depending on the parameterization chosen, but the mode of compaction remains the same over the season. SnowModel simulates high compaction rates for all parameterizations, which is correct for the start of the season but an overestimation after mid-June. These compaction rates implicate changes in the thermal conductivity of the snowpack and thus changes in the melting. The different snow density parameterizations in SNOWPACK are still developed and improved (e.g. Keenan et al., 2021), but an improvement of snow density parameterizations in semi-arid regions shows a demand for snow density measurements, as deriving density from SWE/SD measurements is biased over direct density observations using manual measurements (Smith et al., 2017).

Subsequently, the albedo parameterization appears an important parameter to be properly assessed (Fig. 4, 6), as also reported by the studies performed in Alpine regions (Etchevers et al., 2004; Zolles et al., 2019). This can be surprising at first glance as in the semi-arid Andes the ablation is mainly driven by the sublimation and the albedo parameterization is generally crucial to accurately simulate the melt. However, according to the results presented here, the two models agree with the larger sensitivity to the albedo parameterization. The impact of parameterization choice differs for the two models as the uncertainty is directly related to the difference in snow physical representation and the characteristics of the models. Indeed, the range of the ensemble approach simulated by SNOWPACK is higher than that simulated by SnowModel, which is directly related to both higher number of parameterization possibilities for SNOWPACK and more complex physical representation of the processes.

Likewise, results presented here show that the main sensitivity remains in the forcing uncertainty, in agreement with previous studies (Magnusson et al., 2015; Günther et al., 2019; Raleigh et al., 2015; Schlögl et al., 2016). For instance, Magnusson et al. (2015) found that the models of different complexity (temperature-index models vs. physical models) show similar ability to reproduce daily observed snowpack runoff, and concluded that the forcing uncertainties is the greatest factor affecting model performance, rather than model parameterizations. However, as mentioned by Raleigh et al. (2015), simulated SD and SWE are critically sensitive to the relative magnitude of errors in forcing. Raleigh et al. (2015) and Schlögl et al. (2016) also mentioned that precipitation bias (or correction of the undercatch of the precipitation gauge) was the most important factor, in agreement with the findings of our study.

Finally, Rutter et al. (2009) pointed out that no universal 'best' model exists and model performance directly depends on calibration of the models to the specific study site. Here similar conclusions can be drawn for both Alpine and semi-arid environments, namely that the choice of model structure and parameterizations, along with a specific calibration of the parametrizations for the study site, has a major impact on the performance.

## 5.2 Limitations of the study and further work

The sensitivity study of the two models to the forcing is done by adding a bias to the meteorological variables with ranges derived from literature. It was also possible to add random noise to the data, but this does not necessarily preserve the physical consistency and would lead to low sensitivity of the models (results not shown), as random noise counterbalances each other, which has also been investigated by Raleigh et al. (2015). It would also have been possible to apply a random perturbation (e.g. Charrois et al., 2016) using a first-order auto-regressive model (Deodatis and Shinozuka, 1988). However, the forcing bias does not affect the conclusion of the relative comparison of the two models which only requires the exact same forcing as input to be relevant. For the same reason, the choice of method applied for the forcing correction (i.e. for precipitation) and reconstruction (i.e. for the TA and RH) would not affect the conclusions of the model comparison. However, due to the precipitation uncertainty related to measurements errors, and also because the sensor locations may not be representative of the area, different ways to correct the precipitation data were proposed. Günther et al. (2019) and Grünewald and Lehning (2014) already outlined that the snow cover is spatially heterogeneous even at very small scales due to topographic and microclimatic effects on accumulation, redistribution, and ablation processes, introducing an uncertainty in validation data. We also show that in any case, due to i) the question of the sensor location representativity of the area, ii) the precipitation undercatch because of the wind, and, iii) the high sensitivity of models to precipitation uncertainty, this study highlights the complexity and necessity of accurately measuring precipitation. Additionally, possible corrections also depend on the availability of observations, but this study was restricted to not using SWE and SD as forcing, as these parameters were needed as validation data. Therefore, we chose a precipitation correction that overestimates snowfall at the start of the season, but does not capture the increase of SD and SWE in mid-June, resulting in a good agreement between simulated and observed SWE from the beginning of July (e.g. Fig. 4).

The ensemble approach with different parameterizations is built considering limited parameterization options, contrary to other studies where a large number of physical options are considered (e.g. Essery et al., 2013; Lafaysse et al., 2017; Zolles et al., 2019). In our case, choosing snow models with different physical complexities limits the number of calibration possibilities, as the parameterization of the same variables are chosen for the comparison. Thus, some parameterizations, such as the choice of atmospheric stability correction only available in SNOWPACK, were excluded and this model is calibrated following the same options found in SnowModel (Sect. 3.2.1).

Testing different albedo parameterizations is chosen as i) different options are possible in both models and ii) previous studies concluded that the largest absolute uncertainties originate from the shortwave radiation and the albedo parameterizations (e.g. Zolles et al., 2019). The sensitivity test to different fresh snow density parameterizations was also chosen as previous studies identified this parameterization as a significant uncertainty in model calibration (e.g. Essery et al., 2013). Finally, energy balance models are known to be sensitive to $z_0$, especially in cold and dry regions where sublimation is the main ablation process (e.g. Réveillet et al., 2020). However, due to this important sensitivity and the absence of measurements to properly calibrate this value, two values for $z_0$ were implemented, but this still might underestimate the possible range of $z_0$ values. Otherwise, despite the choice of limiting the parameterization options, SNOWPACK's sensitivity to model parameterizations is

470 evaluated based on 40 simulations, whereas SnowModel's evaluation is based on 12 simulations only. However, the difference of the number of simulations does not impact the conclusion, as the width of the spread of different parameterizations was not quantitatively assessed.

Among the possible settings of the model, the snow transport option has not been activated, while the option is available in both models. However, due to the strong wind speed characteristics of the study area (Gascoin et al., 2013, and Fig. 2), snow

transport is expected to be considerable. Yet, snow transport estimation remains out of the scope of this study, focused on energy balance comparisons, mainly to assess differences in sublimation rates. Also, in a study performed in the Pascua-Lama catchment, a region to the north of Tapado AWS, Gascoin et al. (2013) highlighted that the inclusion of SnowTran3D does not change the fact that the model is unable to capture the small-scale snow depth spatial variability (as captured by in-situ snow depth sensors). Finally, snow transport in SNOWPACK can only be simulated in the 3D domain with SD as forcing, which then

could not be used as validation data. However, due to the importance and complexity in modelling snow transport, properly assessing its impact could be assessed in future work.

## 6 Conclusions

Snow models are key to quantify runoff and provide accurate water availability projections. The aim of this study is to compare

two snow models, SNOWPACK and SnowModel, and evaluate their sensitivity relative to parameterization and forcing. For that purpose, the two models are run over the 2017 snow season, at local point, and forced with i) the most ideal set of input parameters, ii) an ensemble of different physical parameterizations and iii) an ensemble of biased forcing.

The most ideal set of input parameters consisted of observed forcing and the validation parameters (SD and $S_\downarrow$ for SNOW-PACK; SWE for SnowModel) given as input. Hence, the models were able to assimilate the forced precipitation to correct

for undercatch in the precipitation gauge. SNOWPACK is able to approach the observation very well (i.e. min. $RMSE$ of 9.2 cm, max. $R^2$ of 0.93 calculated with the observed and simulated SD), but SnowModel only adjusts the precipitation at two precipitation events, still leading to undercatch. The final correction of the precipitation data was done with an equation based on TA and WS, as it was unwanted to adjust the precipitation with SNOWPACK's assimilated data, as this assimilated data is built from data, which were required for model evaluation. The parameterization simulations were done considering different

parameterizations of the albedo and the fresh snow density and different values for $z_0$. Results indicated a significant difference related mainly to the parameterization choice of the albedo and $z_0$. However, the impact of the albedo affects the two models differently. For SnowModel, the albedo parameterization has a significant impact on the simulated sublimation during the cold period while SNOWPACK simulates similar sublimation rates for all the possible parameterizations. The choice of albedo parameterization in SNOWPACK has direct consequences on melt at the end of the season. The model differences are mainly

related to the model characteristics (e.g. the consideration of the water evaporation and refreezing into the snowpack), and the more complex representation of the snow physics in SNOWPACK. However, the models are both sensitive to the chosen $z_0$, leading to sublimation rates ranging from 36% up to 86% of total ablation.

In addition, results presented in this study highlight a larger uncertainty depending on the model parameterization (despite the limited number of options chosen) than between the two models (despite the significant differences in their physical complexity).

Otherwise, for both models, results show high levels of uncertainty related to forcings which is directly related to the bias chosen, but the spread of the uncertainty for both models is approximately the same. SNOWPACK and SnowModel are highly influenced by precipitation uncertainties. Both models show similar levels of uncertainty, in modelling the end of the season. .

Our study covers the winter season of 2017, and our conclusions on model sensitivity to various parameterizations are specific to that period. In further studies, simulations could be performed over a larger time period, and at distinct places to complement our results. Furthermore, additional models could be used, in particular snow models with similar physical complexity. Such work would provide additional information of the parameterization sensitivity by allowing a comparison based on a larger choice of possible parameterizations.

*Data availability.* Part of the data used in this paper (AWS data) can be accessed at https://www.ceazamet.cl. SnowModel can be accessed by contacting the administrator, Glen E. Liston. SNOWPACK is an Open Source model and can be accessed at https://gitlabext.wsl.ch/snow-models/snowpack. For any other access to the data presented in this study, please contact the authors.

*Author contributions.* AV conducted data preparation, ran the numerical experiments and produced the figures. AV and MR designed the modelling strategy. MR and SM designed the study. All authors contributed to the results analysis and to the preparation of the paper.

*Competing interests.* The authors declare that they have no conflict of interest.

*Acknowledgements.* We thank Michael Lehning and two anonymous referees as well as the editor, Brice Noël, for their constructive comments, which helped to sharpen the scope of this study. We also thank Glen E. Liston for providing the code of SnowModel and are grateful to CEAZAmet and the CEAZA glaciology group for maintaining the Tapado AWS and data centre.

Marion Réveillet and Shelley MacDonell were supported by CONICYT-Programa Regional R16A10003, and the Coquimbo regional government FIC-R(2015) BIP 30403127-0. Marion Réveillet was also supported by the ANR program: ANR-16-CE01-0006 EBONI.

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
