# Peer review of "Snow model comparison to simulate snow depth evolution and sublimation at point scale in the semi-arid Andes of Chile"

_The Cryosphere, 2021_

## Referee Comment (RC2)

Review "Snow model comparison to simulate snow depth evolution and sublimation at point scale in the semi-arid Andes of Chile" by Annelies Voordendaag et al.

By Michael Lehning

General:
The paper investigates two snow models with respect to their sensitivities on input and parameterizations in simulating snow in a dry high-elevation environment. This is in principle a useful exercise given the importance of snow and snow melt as a water resource in these ecosystems. I share the motivation that snow model evaluation is needed for the particularly dry environment in the Andes. The paper is well-written and the presentation of the material is clear. However, I have major concerns about the execution of the study. The main problem is that the main features of the mass balance are not reproduced by neither model despite the "calibration" attempts. Since there is a strong influence of total mass (SWE) and depth of the snow on processes such as sublimation and melt, which are in the focus of the study, the point of departure is insufficient. From the SWE and snow depth curves presented in the paper, I would hypothesize that you have significant snow accumulation and occasional erosion from snow transport at your site during the main winter. Since snow transport very heavily influences snow sublimation and total mass influences melt, the results obtained without reproducing at least approximately the local mass balance appear not trustworthy.
A good mass balance could in principle be simulated with SNOWPACK by using the transport module. As a minimum, I would request that SNOWPACK is used to first generate a best estimate mass input (by using the snow depth forcing feature) and then start the sensitivity analysis.

Another major point is that I cannot see in how your analysis supports your conclusion that parameterization would be more important than model choice or structure. The two striking differences between the two models are 1) the strong settling/melt of SnowModel already during the main winter and 2) the rapid melt-out at the end of the season. These two characteristics are qualitatively not changed by any of the parameterization changes. I would in fact assume that they have to do with model physics (settling law / water transport / refreezing) and model structure (number of layers).

Let me further suggest that the choice of model variants (such as picking the sub-model for albedo) is not called calibration. A calibration is a procedure, by which you determine the value of a free parameter. What you do is not the same as calibration.

Specific Comments:
l. 11: "varies EVERY eight days" is not clear
l. 48: not sure I agree with "ESPECIALLY in regions where sublimation …." You should give a justification here
l. 77: In the picture, I can see some lichen vegetation
l. 85 ff: A good recent paper discussing errors in this type of SWE measurement is [*Gugerli et al.*, 2019].
l. 95/96: This only makes sense if you had included soil layers below the snow.
l. 101: Why using a moist adiabatic lapse rate is such a dry environment? Give a justification!

l. 114: Very small roughness length!

l. 122: Such spatial variability has been investigated by [*Grunewald and Lehning*, 2015]

l. 194 ff: If I understand the text and Eq. (2) correctly, this must produce a value at every time step including negative values. Can you please clarify? From your figure S2.1 this appears not to be the case but then the presentation may be wrong. Please check / clarify.

l. 223 ff: Note that there is a strong cross-sensitivity with settling. Also all of these results will look differently when snow transport is properly taken into account. For the diverse parameterizations, I would emphasize that they are listed and named in S4.

l. 235: See, this is where the mass balance is important: if you had a correct mass balance then you could see which runs do reproduce the melt-out date, which is an important quantity to model.

l. 247: Justify the statement!

l. 303: latent heat is also a turbulent flux in the ABL!

l. 331: Really Richardson number is not used much any more. Almost everybody uses MO similarity with corresponding stability corrections

l. 373: [*Vogeli et al.*, 2016] have demonstrated how one can get to good spatial mass input

l. 350 ff: A very complete and systematic study on input uncertainties (but using a distributed snow model) is [*Schlogl et al.*, 2016]

References:

Grunewald, T., and M. Lehning (2015), Are flat-field snow depth measurements representative? A comparison of selected index sites with areal snow depth measurements at the small catchment scale, *Hydrological Processes*, *29*(7), 1717-1728, doi:10.1002/hyp.10295.

Gugerli, R., N. Salzmann, M. Huss, and D. Desilets (2019), Continuous and autonomous snow water equivalent measurements by a cosmic ray sensor on an alpine glacier, *Cryosphere*, *13*(12), 3413-3434, doi:10.5194/tc-13-3413-2019.

Schlogl, S., C. Marty, M. Bavay, and M. Lehning (2016), Sensitivity of Alpine3D modeled snow cover to modifications in DEM resolution, station coverage and meteorological input quantities, *Environ Modell Softw*, *83*, 387-396, doi:10.1016/j.envsoft.2016.02.017.

Vogeli, C., M. Lehning, N. Wever, and M. Bavay (2016), Scaling Precipitation Input to Spatially Distributed Hydrological Models by Measured Snow Distribution, *Frontiers in Earth Science*, *4*, doi:UNSP 108 10.3389/feart.2016.00108.

---

## Author Comment (AC3)

**Response to review**
Snow model comparison to simulate snow depth evolution and sublimation at point scale in the semi-arid Andes of Chile

Annelies Voordendag, Marion Réveillet,
Shelley MacDonell, Stef Lhermitte

May 5, 2021

Dear editor and reviewers,

First, we would like to thank the editor and reviewers for their careful evaluation of our work and the valuable suggestions, comments and questions. We believe that the manuscript has substantially benefited from the editor's and reviewers' feedback. Below we adress our detailed responses to all the comments.

In this response-to-review document we try to clarify and address each of the suggestions, comments and questions made during the review. Therefore we have copied the comments in blue boxes and have addressed them one by one. In the response we use italic fonts to quote text from the revised manuscript. The revised manuscript will be uploaded soon.

The main changes in the manuscript include a new section presenting an idealised setup to acquire a better precipitation forcing, in response to a main concern of the reviewers. For that purpose, for both models snow depth or SWE have been assimilated to present idealized cases.

Second, the forcing uncertainty have been calculated with a bias, instead of random errors (Sect. 3.3), as initially presented. This choice has been made in agreement with a constructive comment made by a reviewer mentioning that the random errors can counterbalance each other. To avoid the underestimation of the forcing uncertainty, we chose to estimate forcing uncertainty according to Raleigh et al. (2015).

Finally, to better consider the snow roughness value uncertainty, we included a sensitivity to different values in the ensemble simulations in agreement with reviewer comments.

We also want to note that we added Stef Lhermitte as corresponding author.

Yours sincerely, Annelies Voordendag & co-authors

**Response to the Editor B. Noel**

Dear Annelies Voordendag and co-authors,

Thank you for your submission to TCD. As you may know, papers accepted for TCD appear immediately on the web for comment and review. Before publication in TCD, all papers undergo a rapid access review undertaken by the editor with the aim of providing initial quality control. It is not a full review and the key concerns are fit to the journal remit, basic quality issues and sufficient significance, originality and/or novelty to warrant publication. As a result, even a manuscript ranked highly during access review can receive a low ranking during full peer review later. Evaluation criteria are found at: www.the-cryosphere.net. Grades are from 1 (excellent) to 4 (poor).

ORIGINALITY / NOVELTY (1-4): 2
The paper focuses on improving the representation of snow processes at a point location in the Chilean Andes in winter 2017. To that end, the authors estimate the sensitivity of two snow models, namely SNOWPACK and SnowModel, to various parameterizations (i.e., fresh snow density and albedo) and atmospheric forcing perturbations (notably measured precipitation). Models show that sublimation is the main driver of ablation during the snow season (May-November), and that its relative contribution to total ablation (i.e., sublimation ratio) is highly sensitive to the selected albedo parameterization. Atmospheric forcing perturbations also strongly impact model results mainly through precipitation uncertainties. By conducting multiple sensitivity experiments and model evaluation against local in situ measurements, the authors provide valuable insights on model performance, and response to different parameterizations and forcing perturbations.

SCIENTIFIC QUALITY / RIGOR (1-4): 1
The authors conduct a large set of sensitivity experiments, and evaluate model results using three measured variables, i.e., snow depth, snow water equivalent and surface albedo. This evaluation work yields valuable insights on model performance, model differences (i.e., in terms of physical complexity), and response to different parameterizations and forcing perturbations. The authors refer to relevant literature and provide sufficient information on the two selected models, evaluation data sets, calibration methods and design of the sensitivity experiments.

SIGNIFICANCE / IMPACT (1-4): 2
In its current form, the manuscript does not clearly stress the motivations and objectives of the study, notably in the abstract and conclusions. For instance, conclusions remain unclear about e.g. which model parameterizations/forcing perturbations have the strongest impact on modelled ablation components, and thus snow depth evolution in the Andes.

PRESENTATION QUALITY (1-4): 1
The manuscript figures and tables well support the study, results and discussion. The paper could benefit from stylistic improvements and some clarifications.
In brief, this is an interesting and generally well-written modelling study. The editor invites the authors to better stress the motivations and objectives in the abstract and conclusions. This can be addressed in the first round of revisions. For now, the editor recommends publication in TCD.
Kind regards, Dr. Brice Noël

The authors thank the editor for his very positive feedbacks and encouraging remarks. As described in the introduction of this document, we made some changes to the study and as sugfusted by the editor, we better stress the motivations and objectives in the abstract and conclusions.

**Response to the anonymous reviewer #1**

**Main comments**

R1-1: In this manuscript the authors present a sensitivity analysis of two commonly used snowmodels, SNOW-PACK and Snowmodel, for a semi-arid Andes catchment. The authors aim to quantify the impact of various parameter and parameterization selections and impact of forcing uncertainty on a sublimation dominated catchment. Overall, this manuscript is clear, well written, and provides useful results. Research topics like this have a tendency to end up very location-centric and not widely applicable to the larger community. However, I do not believe that is the case here. There are sufficient linkages with existing work.

The authors thanks the reviewer for its careful evaluation of our work and the valuable suggestions, comments and questions. .

R1-2: I have two concerns: One, the description of precipitation measurements is unclear to me. Figure 1 suggests an unshielded Geonor gauge is used. However, the authors use an Alter-shielded correction factor. This should be clarified in the text. If an unshielded gauge was indeed used, then a) a different factor should be used and b) the uncertainty in the precipitation is massive and I am then not completely convinced. I also note that MacDonald (2007) is a grey-literature source (conference proceedings), and I am curious as to why the authors chose this correction versus some of the "more standard" WMO/Goodison corrections?

First, the description of precipitation measurements have been clarified. For instance, in the measurement description you can now read: *This gauge is an unshielted, unheated weighing bucket precipitation gauge filled with anti-freeze liquid and oil to prevent freezing and evaporation respectively.*
Second, in agreement with your comment and comments made by the other reviewers, we:
a) forced the models with snow depth or SWE to assimilate precipitation data that lead to more realistic results

and we therefore add a specific and new section 'Idealized setup', to present these results.

b) we corrected measured precipitation with an other approach (i.e.Wolff et al. (2015)) which leads to an amount of precipitation closer to precipitation reconstructed from SWE.

To clarify the different precipitation used in this study, a figure presenting the diffent options has been added in the revised version.

> R1-3: Two, it seems to me the authors are using only the instrument measurement uncertainty. This should be noted in the text. However, I'm surprised the authors did not use the uncertainty ranges and distributions from Raleigh et al. (2015, Table 3) which include more 'real-world' uncertainty ranges. I believe the manuscript would benefit from using these distributions and ranges. I believe this would more cleanly link this work with existing studies and increase the contribution.

We agree with the reviewer that considering only the measurement uncertainty might underestimate the forcing uncertainty. We therefore apply a bias to the forcing data, according to the ranges described in Raleigh et al. (2015, Table 3) in the revised version of the manuscript. In addition, as the method chosen in the initial version (i.e. random errors) tended to counterbalance each other, we decided to only apply a bias in the revised manuscript. Furthermore, the TA and RH are assumed to already experience some random noise due to its interpolation.

**Detailed notes**

> R1-4: L18 "complexities" should be elaborated on as to what the authors mean by this, as everyone has different definitions.
>
> R1-5: L18 "physical approaches" do you mean "physically-based" or "physics-based" here? All approaches should be physical.

We clarified the text. You can now read:

*Several models, with different complexities in the representation of different snow processes, from empirical to physically-based approaches, (..)*

> R1-6: L22 "These approaches, coupled with snow models" I don't understand what you mean by this.

We agree that this statement was unclear. We meant to say that our approach is the use of the energy balance equation. The text now reads:

*The use of the energy balance equation, coupled with snow models, enables (..)*

> R1-7: L32 Should cite Essery (2015) as well

Done.

*In addition to the development of new models, many studies have focused on model improvements offering different parameterizations in a single model (e.g. Douville et al., 1995; Dutra et al., 2010; Essery, 2015).*

> R1-8: L53 "this" refers to what?

"This" refers to the importance of snow as water resource. This in now clarified in the revised version:

*Despite the importance of snow as water resource, quantifying (..)*

> R1-9: L54 I would also note dry air (important for sublimation)

We agree with your statement and add this information in the revised manuscript:

*i) high sublimation rates related to strong solar radiation, cold air temperatures, arid atmosphere, and high wind speeds*

> R1-10: L65 1000 seems arbitrary, perhaps note as much or describe why this number was chosen

This was indeed an arbitrary amount and a compromise between computation effort and reliable results. We have added the following:

*We have chosen 1000 runs as a compromise between computational effort and a reliable confidence interval.*

**R1-11: L72 "to assess the sensitivity" of what?**

We clarified this phrase:
*To assess the sensitivity of the models to the representation of snow physics and meteorological forcing, we (..)*

**R1-12: L72 "a permanent station" clarify this is a meteorological station**

We clarified this: *a permanent meteorological tower since 2009*

**R1-13: L80 If this table can be included in the main text, I think you should do so**

We agree with the reviewer and have added this table to the main text in the revised manuscript.

**R1-14: L80 Describe shielding for Geonor here.**

We extended the text with:
*This gauge is an unshielted, unheated weighing bucket precipitation gauge filled with anti-freeze liquid and oil to prevent freezing and evaporation respectively.*

**R1-15: L96 "Physical equilibrium" what does this mean?**

With physical equilibrium we mean the realistic climatology for the model (e.g. surface temperature) after the model is being started from other initial conditions, but in the revised manuscript this entire phrase will be removed.

**R1-16: "June 11:00 and 31 Oct" replace 'and' with 'to'**

Done.
*(23 June 11:00 to 31 October 10:00 due to sensor failure)*

**R1-17: L110 describe shielding**

We solved this in comment R1-14.

**R1-18: L118 Blowing snow sublimation will also lower on ground swe and should be noted as this further adds uncertainty to the reconstruction.**

We agree with the reviewer that sublimation influences the amount of SWE and we therefore reconstructed the precipitation from SWE at moments that also precipitation was registered at the precipitation gauge to ensure that the change in SWE is caused by precipitation.

**R1-19: L165 add "snow" roughness**

Done.
*Atmospheric stability and snow roughness length ($z_0$)*

**R1-20: L165 I'm surprised the roughness length is so small. This is well on the lower end of what is reported in the literature. It seems to me to be calibrating sensitivity to the turbulent fluxes, suggesting that they are being overestimated with more 'reasonable' $z_0$ values. Could this be related to the stability parameterization being insufficient in this area?**

We agree with your comment. However as no measurements are available to better calibrated this value we chose to consider different values in the evaluation of the model sensitivity to the parameterization choice (i.e. $z_0 = 1\,\mathrm{mm}$ as in the intial version and $z_0 = 1\,\mathrm{cm}$). We also better discuss this point in the discussion section of the revised manuscript:
*The turbulent fluxes parameterization is sensitive to the roughness value and observations, such as Eddy Covariance measurements, are essential to accurately parameterize the turbulent fluxes (e.g. Conway and Cullen, 2013; Litt et al., 2017; Réveillet et al., 2018). Other values in literature at other locations also show that the snow roughness length vary widely and spans two orders of magnitude (Gromke et al., 2011; Poggi, 1977; Bintanja and Broeke, 1995; Andreas et al., 2005). Due to the absence of such measurements, and also due to the strong variability of this*

*value over the time (e.g. MacDonell et al., 2013b; Pellicciotti et al., 2005; Nicholson et al., 2016), it was decided to use two different values for $z_0$ (1 mm and 1 cm) and similar sensitivity ranges for SNOWPACK and SnowModel were found (e.g. Fig. 5.)*

> R1-21: L230 all RMSE values need a unit, even if it's (-) for albedo.

Done and also solved this in the rest of the document.

> R1-22: L230 "compared to the albedo" what is compared ?

We calculated the *RMSE* and $R^2$ between the modelled and the observed albedo.
*(i.e. RMSE of 0.09 (-) and $R^2$ of 0.85 calculated with the observed and simulated albedo)*

> R1-23: L241 rmse units

Done.

> R1-24: Figure 3: Legend should be added. Albedo yaxis needs more ticks, at least in the 0.5-1 range.

We added extra ticks to the y-axis and the legend has been added. See Fig. 1 in this document.

> R1-25: L321 remove "total"
> R1-26: L321 add "in other alpine"

Done.
*This conclusion, found here in an arid environment, is in agreement with the studies performed in other alpine areas.*

> R1-27:L351 fix reference ( )

Done.

**Response to the reviewer #2: Michael Lehning**

**Main comments**

> R2-1: The paper investigates two snow models with respect to their sensitivities on input and parameterizations in simulating snow in a dry high-elevation environment. This is in principle an useful exercise given the importance of snow and snow melt as a water resource in these ecosystems. I share the motivation that snow model evaluation is needed for the particularly dry environment in the Andes. The paper is well-written and the presentation of the material is clear.

We are very thankful for the constructive and positive comments of reviewer #2.

> R2-2: However, I have major concerns about the execution of the study. The main problem is that the main features of the mass balance are not reproduced by neither model despite the "calibration" attempts. Since there is a strong influence of total mass (SWE) and depth of the snow on processes such as sublimation and melt, which are in the focus of the study, the point of departure is insufficient. From the SWE and snow depth curves presented in the paper, I would hypothesize that you have significant snow accumulation and occasional erosion from snow transport at your site during the main winter. Since snow transport very heavily influences snow sublimation and total mass influences melt, the results obtained without reproducing at least approximately the local mass balance appear not trustworthy. A good mass balance could in principle be simulated with SNOWPACK by using the transport module. As a minimum, I would request that SNOWPACK is used to first generate a best estimate mass input (by using the snow depth forcing feature) and then start the sensitivity analysis.

We totally agree with the underestimation of the precipitation input. In agreement with this statement and also

comments made by the other reviewers, we added a new section *Idealised setup*, where we use both SNOWPACK and SnowModel to assimilate a more 'realistic' precipitation set. In addition, it raised the awareness that we needed another correction for the precipitation. As also mentioned in response to reviewer 1, we therefor chose an other method (based on the study by Wolff et al. (2015)) to correct the measured precipitation which better corresponds to the total SWE. In the revised version the simulations with parameterization ensembles and forcing ensembles are therefore performed with a more appropriate precipitation amount. The parameterizations ensembles are shown in Fig. 1 of this document.

[Figure]

Figure 1: a-b) SD, c-d) SWE and e-f) albedo simulations (coloured) of the ensemble approaches for SNOWPACK (red) and SnowModel (blue) and observations (black). The bold coloured lines show the reference simulations chosen as the most optimal parametrizaton set according to the measured albedo. The dotted line of SWE indicates the less reliable (lower) SWE measurement from thallium rays.

Finally, in the revised version, the snow transport in SNOWPACK was activated. However, as simulations are performed at point scale, the impact on snow transport is poor, likely related to the complexity of snow transport (erosion/depostion) at point scale. Further studies at larger scale would be very interesting, especially in such windy areas, but this is out of the scope of the present work. This is however discussed in the discussion of the revised manuscript.

> R2-3: Another major point is that I cannot see in how your analysis supports your conclusion that parameterization would be more important than model choice or structure. The two striking differences between the two models are 1) the strong settling/melt of SnowModel already during the main winter and 2) the rapid melt-out at the end of the season. These two characteristics are qualitatively not changed by any of the parameterization changes. I would in fact assume that they have to do with model physics (settling law / water transport / refreezing) and model structure (number of layers).

According to your comment we decided to compare and discuss independently the model parameterization uncertainties and the model structure/physics differences. For that purpose, first the model parameterization choice is quantified for both models (Fig. 1 of this document). Then in the discussion we elaborate the impact of the differences of the model physics and structure (i.e. number of layers, refreezing, water transport) according to your

suggestion.

> R2-4: Let me further suggest that the choice of model variants (such as picking the sub-model for albedo) is not called calibration. A calibration is a procedure, by which you determine the value of a free parameter. What you do is not the same as calibration.

Thank you for this clarification. For the specific case of the albedo, it is now mentioned that different sub-models for the albedo are tested. In addition, parameterization is generally preferred throughout the manuscript instead of 'calibration'.

**Specific Comments**

> R2-5: l. 11: "varies EVERY eight days" is not clear

This section had been removed from the abstract, as the scope of the study slightly changed.

> R2-6: l. 48: not sure I agree with "ESPECIALLY in regions where sublimation ...." You should give a justification here.

By 'especially' we meant that not a lot of snow models have been applied to semi-arid regions. We modified the sentence in agreement with your statement:
*... in particular in regions where sublimation is the main ablation process, due to the lack of snow modelling studies in semi-arid regions (Gascoin et al., 2013; Réveillet et al., 2020; MacDonell et al., 2013a; Mengual Henríquez, 2017).*

> R2-7: l. 77: In the picture, I can see some lichen vegetation

We adjusted the text:
*At this elevation, vegetation is extremely sparse.* .

> R2-8: l. 85 ff: A good recent paper discussing errors in this type of SWE measurement is Gugerli et al. (2019).

Gugerli et al. (2019) used a cosmic ray sensor mounted on the glacier surface. While this paper provides valuable information about the uncertainty of this sensor in a glaciarized area, we are not sure about the relevance in comparing this uncertainty to our study, as in our case a different sensor was used on snow surface.

> R2-9: l. 95/96: This only makes sense if you had included soil layers below the snow.

The text now reads:
*The period between 5 May and 30 Nov 2017 has been covered to model the snow evolution in the austral winter.*

> R2-10: l. 101: Why using a moist adiabatic lapse rate is such a dry environment? Give a justification!

We agree that using a moist adiabatic lapse rate migh not be appropriate in dry environment. However, in our study the lapse rase wasn't a moist adiabatic one. It has been calculated with the available data. This point point have been clarified in the revised manuscript as follows:
*For TA, a daily temperature lapse rate (Blandford et al., 2008) was calculated using TA measured at La Laguna and Paso Agua Negra AWSs (1565 m elevation difference) between 2014 and 2017.*

> R2-11: l. 114: Very small roughness length!

In agreement with your comment a roughness length of 1 cm has been chosen. We added this to the Sect. 4.1:
*Eq. (12) from Wolff et al. (2015) with WS corrected to gauge height using a logarithmic wind profile (e.g. Lehning et al., 2002) and a $z_0$ of 0.01 m is used as precipitation data in the further study, (..)*
However, we made a sensitivity test on this value and does not lead to significant difference in the precipitation correction.

> R2-12: l. 122: Such spatial variability has been investigated by [Grunewald and Lehning, 2015]

We moved this phrase to the section *Idealised setup* and added the reference.
*Preliminary results showed simulated SWE and SD to be more than two times lower than the observed SWE. This is caused by an underestimation of the precipitation measurements, as the AWS is placed in a concave area that collects more snow than the Geonor precipitation gauge. This is in correspondence with research by Grünewald and Lehning (2014) on the spatial variability of SD measurements.*

> R2-13: l. 194 ff: If I understand the text and Eq. (2) correctly, this must produce a value at every time step including negative values. Can you please clarify? From your figure S2.1 this appears not to be the case but then the presentation may be wrong. Please check / clarify.

According to comments made by reviewer #1, the method to calculate the precipitation uncertainty has been modified. We now use a uniformly distributed bias for the precipitation ranging between positive values. In case of the other forcing variables, a normally distibuted bias has been applied (and reported in Table 2) and if negative values (e.g. for RH, WS, $S_\downarrow$) occurred here, the value was set to 0 (or 0.1 m s$^{-1}$ for WS). This is now clarified in the revised version:
*To assess the model sensitivity to meteorological measurement uncertainties, a bias has been applied to the meteorological forcing presented in Sect. 2.2 to generate an ensemble of 1000 forcing files. Raleigh et al. (2015) have shown that the model outputs are more sensitive to forcing biases than random errors. Therefore, all input variables except P were modified by adding hourly biases with a normal distribution $N(\mu = 0, \sigma^2)$ with $\sigma$ the uncertainty range taken from Raleigh et al. (2015) and reported in Table 2. The biases has been kept in the range (Table 2) by assuming that the 99.7% of the bias, thus $3\sigma$, is within this range. This positive component of the range is divided by three and multiplied with a normal distributed random number and added to the observed forcing. We have chosen 1000 runs as a compromise between computational effort and a reliable confidence interval.*

> R2-14: l. 223 ff: Note that there is a strong cross-sensitivity with settling. Also all of these results will look differently when snow transport is properly taken into account. For the diverse parameterizations, I would emphasize that they are listed and named in S4.

We agree with your comment. However, the aim of the study is to compare the sensitivity of both models to diverse parameterizations. As snow transport cannot be activated in SnowModel at point scale, we decided to make the comparison with SNOWPACK without the snow transport. We aware that this is a limitation, and this is therefore discussed in the revised version.

Otherwise, in agreement with your main comment, the snow transport has been activated for the idealized case. As mention above, as simulations are performed at point scale, the impact on snow transport is poor, likely related to the complexity of snow transport (erosion/depostion) at point scale. Note also that in the revised version, we referred again to the equations and references of the parameterizations that can be found in the supplementary material.

> R2-15: l. 235: See, this is where the mass balance is important: if you had a correct mass balance then you could see which runs do reproduce the melt-out date, which is an important quantity to model.

We agree with your statement. Therefore in agreement with your main comments and comments made by the other reviewers, we run an 'Idealised setup', and better corrected the precipitation (please refer to answer to comment R1-2 for more details), to obtain a more realistic mass balance. The figure has been modified accordingly in the manuscript. In addition, the comparison between the observed and simulated SD and SWE has been adjusted and better described in the revised version. This allows a better model comparison.

> R2-16: l. 247: Justify the statement!

We clarified this:
*Where SnowModel relies on two albedo models based on TA and albedo ranges, SNOWPACK relies on empirical relations calibrated with measurements in Switterland and not adapted to the arid Tapado climate.*

> R2-17: l. 303: latent heat is also a turbulent flux in the ABL!

We totaly agree with your comment. However according to the comments made by the 3 reviewers, we decided to

restructure the manuscript and the energy balance section is no longer in the main manuscript.

> R2-18: l. 331: Really Richardson number is not used much any more. Almost everybody uses MO similarity with corresponding stability corrections.

We agree with your statement. However, as the aim of the study was to compare the two models and as SnowModel only offers the correction based on the Richardson number, this correction has been prefered. This is clarified in the revised version:
*The representation of turbulent fluxes in snow models is commonly based on the bulk method, and the atmosphere stability corrections are made based on different possible appraoches such as the the Monin–Obukhov similarity theory or the Richardson number (e.g. Liston and Hall, 1995; Vionnet et al., 2012). In this study, the method based on the Richardson number, was preferred, as both models offered this option, while it might be not the most common one used.*

> R2-19: l. 373: Vögeli et al. (2016) have demonstrated how one can get to good spatial mass input.

Thanks for mentioning this study added. It is a very good example of a correction of spatial mass input. Nevertheless, it would not have been possible in our study area as they used Airborne Digital Sensors (ADS) to scale precipitation input data. We now also use the approach described in the comment to R1-2.

> R2-20: l. 350 ff: A very complete and systematic study on input uncertainties (but using a distributed snow model) is Schlögl et al. (2016).

We added this reference to the text:
*Likewise, results presented here show that the main sensitivity remains in the forcing uncertainty, in agreement with previous studies (Magnusson et al., 2015; Günther et al., 2019; Raleigh et al., 2015; Schlögl et al., 2016).*

**Response to the anonymous reviewer #3**

**Main comments**

> R3-1: The manuscript "Snow model comparison to simulate snow depth evolution and sublimation at point scale in the semi-arid Andes of Chile" by Annelies Voordendag et al. discusses the application of two snow models at a site with an automatic weather station in a dry mountainous area in the Chilean Andes. The snow models provide mass balance components, which is of importance for investigating water resources in such areas. The model simulations suggest a very strong sublimation flux to the atmosphere, depleting significant amount of snow mass on the ground. By perturbing model settings and forcing data, uncertainties are quantified. Generally, the paper is well written and the topic is highly relevant, and the approach by including two snow models and doing the sensitivity study by perturbing model settings and forcing data is very solid. This makes for excellent ingredients for the manuscript.

We would like to thank the reviewer for his/her careful evaluation of our work and the valuable feedback and the very positive comments.

> R3-2: However, it is surprising to see how poorly the models are able to reproduce the snow cover at the site. This deserves some more attention, because in my opinion, the agreement is so poor, that the trust I have in the simulated mass balance components is also severely limited. Intuitively when looking at the results and the discussion by the authors, a problem could be that the place is so wind affected, that the models underestimate density. That could explain the much stronger settling in the simulations than observed. As the authors discuss themselves, snow density is an important factor for sublimation, since the snow surface temperature depends on it. So I wonder if it is maybe a better approach to run the models with a fixed fresh snow density of let's say 350 kg m$^{-3}$, to see if the agreement improves? Or also to analyze the combined SWE increase and snow depth increase to get an estimate of the fresh snow density? This value could be compared to the parameterizations from SNOWPACK and SnowModel (add observed density from SWE/SD to Fig. S6.1 for example). It is definitely an aspect that is not sufficiently discussed in the current manuscript where the discrepancies between models and observations come from. It's an almost more interesting aspect of the

study that the models are apparently very poorly able to capture the processes at this site.

We agree that the models very poorly simulated the SD and SWE in the initial version. This was principally related to the strong underestimation of the mass as both simulated SWE and SD where understimated in the initial version. Therefore, in agreement with your comments (R3-2 and R3-3), but also with the comments made by the two other reviewers, we adjusted the mass with a better precipitation correction. Despite this adjustment, the SD is still underestimated at the end of the season likely related to a too high snow density as this underestimation is not observed for the SWE (see Fig. 1 in response to R2-2). As no density measurements were available, the fresh snow density parameterization was simulated using the 5 different parameterizations available in SNOWPACK and 3 options for SnowModel. These parameterizations allows to consider the snow density evolution during the season as illustrated by Fig. 2. The main bias occurs during the second part of the season and didn't find an adjustment that fits over the entire season. The option of a fixed density will also provide good estimates only for a small part of the season. However we fully agree that this bias is a limitation, especially because of the impact on the snow surface temperature and therefore to the turbulent fluxes and sublimation rates. This point is discussed in the discussion of the revised manuscript.

[Figure]

Figure 2: Snow density simulations of the complete snowpack for a) 40 calibrations for SNOWPACK (red) and b) 12 calibrations for SnowModel. The reference runs are bold and we also give the $\rho_{snow}$ calculated with SD and SWE (black). Snow density measurements were unavailable.

R3-3: I'm also somewhat confused that the total SWE in the models is so much underestimated. The models roughly produce ~175 mm w.e. in sublimation/evaporation and ~100 mm w.e. runoff. So the total mass input to the models (around ~275 mm w.e.) is less than the maximum SWE observed at the site (300-350 mm w.e.).

> The maximum SWE is about half in the simulations than what is observed. In L129-123, authors discuss that a precipitation reconstruction based on the SWE time series overpredicts total mass, but I don't find any compelling reason presented why an underprediction from the precipitation gauge is preferred over an overestimation from the SWE reconstruction. Furthermore, I think generally undercatch corrections vary so much, and are often found to be site-specific and setup-specific, that I think authors should take some freedom to improve the undercatch correction for this specific setup.

We agree that this underestimation was a limitation in the first version of the manuscript. In agreement with your comments and the comments made by the other reviewers we first add a specific section of an idealised case. For that purpose the simulation used assimilated SD or SWE to reconstruct the precipitation. However, as such measurements are not always avalable and as by using it as input, it cannot be used as validation, we run other simulations based on corrected precipitation in the further study.

   Following your comment, we tried different options to correct precitation and also made a comparison between simulations if a precipitation set reconstruted from SWE (PSWE) was used as input. The precipitation correction chosen is a total of 423 mm w.e. compared to 477 mm w.e. at the end of season for the PSWE, which is feasible, given that PSWE might be overestimated due to some snow drift at particular dates.

**Specific comments**

> R3-4: L255-258: this is not a correct description of how the default version of SNOWPACK works. Unless the authors modified the source code specifically regarding this, SNOWPACK will only use air temperature to distinguish between rainfall and snowfall when driven by a precipitation time series. This wouldn't alter the SWE response, unless runoff occurs. The other criteria are only used when SNOWPACK is driven by a snow height time series.

Thank you for your constructive comment. In response to comments made by the other reviewers the method to calculate the forcing uncertainty has changed and this senstence is no longer in the revised manuscript.

> R3-5: The introduction may need citation of the recent SnowMIP study, with some recent publications: Krinner et al. (2018); Menard et al. (2021)

We added these interesting studies in the revised version of the manuscript:
*Furthermore, the Earth System Model - Snow Model Intercomparison Project (ESM-SnowMIP) compared several snow models to improve the models in the context of local- and global scale modelling (Krinner et al., 2018) and indicated scientific and human errors in snow model intercomparisons (Menard et al., 2021), but the study sites did not include semi-arid regions.*

> R3-6: Section 2.2: please specify if the rain gauge was heated or not.

The rain gauge was not heated. This as been clarified in the revised manuscript:
*This gauge is an unshielted, unheated weighing bucket precipitation gauge filled with anti-freeze liquid and oil to prevent freezing and evaporation respectively.*

> R3-7: L97: I assume that the full period between 23 June 11:00 and 31 October 10:00 is missing for TA and RH, and not only those two specific times.

Indeed, the entire period was missing. This has been clarified as follows: *(23 June 11:00 to 31 October 10:00 due to sensor failure)*

> R3-8: Why was only one year studied while the measurement site has operated for a much longer period (installed in 2009 apparently)? Please also discuss to what extend this year 2017 is representative for the climate at the site (i.e., compare with the full data set in terms of temperature, precipitation and wind speed).

2017 was chosen as a SWE sensor was available for this year. This is now mentioned in the revised manuscript. In addition, 2017 wasn't affected by an ENSO event, and can therefore be considered as neutral (despite quite high temperature). The information of the representativity of 2017 for the climate at the site has been specified in the revised version.

---

## Author Response (AR2)

**Response to review #2**
Snow model comparison to simulate snow depth evolution and sublimation at point scale in the semi-arid Andes of Chile

Annelies Voordendag, Marion Réveillet,
Shelley MacDonell, Stef Lhermitte

July 22, 2021

Dear editor and reviewer,

First, we would like to thank the editor and the reviewer for their careful second evaluation of our work and the detailed suggestions and comments. Below we adress our detailed responses to all the comments.

As for the first round of reviews, in this response-to-review document we try to clarify and address each of the suggestions, comments and questions made during the review. Therefore we have copied the comments in blue boxes and have addressed them one by one. In the response we use italic fonts to quote text from the revised manuscript. Additional to the revised manuscript, we have uploaded a supplementary version of the manuscript with highlighted track changes that indicate where the manuscript has changed (red=removed; blue=added).

Yours sincerely, Annelies Voordendag & co-authors

**Response to the Editor B. Noel**

Dear Annelies Voordendag and co-authors,

We have now received comments from reviewer #3. The reviewer is generally satisfied with your edits and requests minor revisions. The authors should give particular attention to the reviewer's main comments, i.e., clarify which (set of) parameterizations are considered optimal/ideal, and elaborate on the difference between the SWE curves in Figs. 3c and 4c. The authors should also consider improving Fig. 3 (legend/caption) following the reviewer's suggestions. Besides other clarifications requested by the reviewer, the authors can find some additional editor comments/suggestions below.

Based on the above, the editor invites the authors to submit a revised manuscript. Note that the revised manuscript will be re-evaluated by the editor before acceptance for publication in TC.

Best wishes, Brice Noël

The authors thank the editor for his positive feedback and suggestions. As requested, in this revised version of the manuscript, we clarified the set of parameterizations chosen for the optimal case (please refer to the answers to main comments made by the reviewer). We also explained the differences in the SWE curves in figure 3 and 4 (see comment X).

**Editor comments**

As a general comment, the authors should rather use the term "bias" for comparison between models and observations, and "difference" between the two models. For instance, in L307 and L318, "bias" is preferred over "differences" when comparing models with observations.

We agree with the editor, but also note that we prefer to use differences if we compare different parametrizations within one model. L307 and L318 have been adjusted.

L90: Add a reference to Fig. 1b after "(AWS)".

Done.

L177: The sentence is unclear, notably the "resulting which would result in", could the authors reformulate?

The text now reads:
*This cumulative SWE approach reduced the inclusion of deposition caused by snow drift which would have resulted in an overestimation of SWE.*

L186-187: Do the authors mean "The accumulated daily precipitation (P) that agrees best with observed SWE and SD is further used in this study."? The term "P" is defined later in Table 3, but should be first defined here.

We have added "$P_{cor}$" to Fig. 3 and clarified the sentence:
*The precipitation data set ($P_{cor}$) that leads to a simulation with best correspondence to the observed SWE and SD is used in the further study.*

L246: Do the authors mean "1.6 times larger than the observed . . . and the agreement between modelled and observed SD is better for SnowModel than for SNOWPACK . . ."?

We clarified the text:
*The assimilated precipitation is approx. 1.6 times larger than the observed precipitation and the agreement between modelled and observed SD is better for SNOWPACK than for SnowModel (i.e. (..)).*

L262-262: Do the authors mean "Evaluation of simulated SD and SWE based on various parameterizations shows that both models are in good agreement with observations (Fig. 4), although they overestimate SWE at the beginning of the season."?

Yes, the text has been adjusted accordingly:
*Evaluation of simulated SD and SWE based on various parameterizations shows that both models are in good agreement with observations (Fig. 4), although they overestimate SWE at the beginning of the season (May/June).*

L321: Could the authors add the number of days after 30 November?

We cannot add the numbers of days after 30 November, as it was decided to run the model until 30 November. The decision was taken as the measured snow free surface date was already on 16 October and an additional 45 days period after this was assumed to be long enough for the simulations. If we extrapolate the data from Fig. 5b,d, it will be approx. 2/3 days until the red curve reaches 0 cm/mm w.e. but this is just an estimation. To avoid confusion the sentence has been re-written as follows: *(..); for SnowModel this date ranges between 20 August and 29 November (i.e. 101 days) and the range is similar but a bit later in the season for SNOWPACK (i.e. between 29 August and early December).*

L413-414: The sentence is unclear, do the authors mean that similar conclusions can be drawn for both Alpine and semi-arid environments, i.e., the choice of model structure and parameterizations has a major impact on the performance? Please, reformulate.

Yes, we therefore reformulated the text:
*Finally, Rutter et al. (2009) pointed out that no universal 'best' model exists and model performance directly depends on calibration of the models to the specific study site. Here similar conclusions can be drawn for both Alpine and semi-arid environments, namely that the choice of model structure and parameterizations, along with a specific calibration of the parametrizations for the study site, has a major impact on the performance.*

**Editor's suggestions**

L35: to remove "often".

Done.

> L48: "impact of" instead of "consequence of".

Done.

> L68: "based on" instead of "in function of".

Done.

> L82: "falls as snow" instead of "arrives as snow".

Done.

> L107: "Therefore, TA and RH data were ...".

Done.

> L146: "Finally, the three-dimensional model SnowTran3D (Liston and Sturm, 1998), which simulates snow erosion and deposition, is not activated in this study; ...".

Done.

> L221: "perturbed" instead of "disturbed".

Done.

> L255: "markedly increases".

Done.

> L275: "decreases" instead of "reduces".

Done.

> L279: to split the sentence in two "after August. This is likely caused by higher ... or by a bias ...".

Done.

> L290: "12 May, i.e., the first snowfall event, while ...".

Done. The text now reads:
*The mean SD difference between the parameterizations is 20 cm (i.e. 18% of the total SD), with a maximum of 152 cm at the first snowfall event (12 May), while (..)*

> L302: "TA being above the freezing point ... October, resulting in fast melt simulated ...".

Done.

> L402: "is larger than that ... SnowModel, which is directly related to ...".

Done.

> L431-434: "Therefore, we chose a precipitation correction that overestimates snowfall at the start of the season, but does not capture the increase of SD and SWE in mid-June, resulting in a good agreement between simulated and observed SWE from the beginning of July...".

Done.

> L472-473: "assimilated data, which were required for model evaluation.".

The text now reads:
*The final correction of the precipitation data was done with an equation based on TA and WS, as it was unwanted to adjust the precipitation with SNOWPACK's assimilated data, as this assimilated data is built from data, which*

*were required for model evaluation.*

L481: "from 36% up to 86% of total ablation.".

Done.

L483: "despite" instead of "even though".

Done.

L489-491: "Our study covers the winter season of 2017, and our conclusions on model sensitivity to various parameterizations are specific to that period. ... distinct places to complement our results. Furthermore, additional models could be used, in particular snow models with similar ..."

Done:
*Our study covers the winter season of 2017, and our conclusions on model sensitivity to various parameterizations are specific to that period. In further studies, simulations could be performed over a larger time period, and at distinct places to complement our results. Furthermore, additional models could be used, in particular snow models with similar physical complexity.*

Figure S2.1 in L5 of the caption: "represents" instead of "coincides for".

We clarified the sentence:
*PSWE is equal for $z_0$ is 1 mm and 1 cm and thus only the red line is visible.*

Figure S3.1: mention in the caption what K and TI mean.

The caption now reads:
*Observed cumulative precipitation, PSWE and precipitation corrections (MacDonald and Pomeroy, 2007; Smith, 2007; Wolff et al., 2015). The two SWE observations with potassium (K) and thallium (Tl) gamma rays are also given.*

The editor noted the following typos:

- L9: "directly".

- L90: "unshielded".

- L138: "MicroMet is a ...".

- L190: "sensitivity tests".

- L236: "z0 of 1 cm and 1 mm are displayed".

- L299: "Switzerland".

- L365: "one of the most".

- Table 2: "1 default option and 2 from SNOWPACK".

- Fig 5: For consistency, "SNOWPACK" instead of "SnowPack" in the caption.

All typos have been corrected.

**Response to the anonymous reviewer #3**

**Main comments**

The authors provided a thoroughly revised manuscript, with most issues properly addressed. There are a few remaining issues though, which I think the authors should take into consideration before the manuscript can

be accepted for publication. Please find those below.

We would like to thank the reviewer for his/her second evaluation of our work, the valuable feedback and the positive comments.

> The most important issues are with the new section "Idealised simulations". First, I'm not sure if idealised is the best term, since it mainly concerns the reconstruction of precipitation from SD and/or SWE. So maybe the title should reflect that. The main confusing part for me here is what is considered optimal or idealised. In Fig. 4, the thick line indicates the most optimal settings. I noticed in that figure for the SNOWPACK results that the drop in SWE towards the end of September in Fig. 4c is much smaller than in Fig. 3c. So there are clearly other settings being used, and it is unclear to me how the results can be so different when the manuscript suggests that both figures are created with a kind of optimal/best settings. It has an impact on the precipitation reconstruction from snow depth, since SNOWPACK will add precipitation to match the observed snow depth after this drop that occurred at the end of September.

The term "idealised" was chosen as the most "perfect" set of input variables was used as input. This includes indeed a precipitation assimilation, which has a significant (and main) effect on the simulated snow depth and SWE. But it also includes other important variables such as the observed albedo (for SNOWPACK). Indeed, this is the cause of the bigger drop of the SWE (and SD) at the end of September in Fig. 4. In Fig. 4 only the $S_\downarrow$ was used and the albedo is simulated with the available parametrizations in SNOWPACK. During field observations we have seen that the patchy snow areas sometimes get covered with dust, probably causing the drop in observed albedo and thus the faster melt. This can, of course, not be modelled with the implemented parametrizations. The term "idealised" is therefore chosen and corresponds to all the "best" input data possible. However, first, all this data are not always available and second, it is not possible to use SD or SWE and observed albedo as validation data, if it is also used as input.

We therefore consider and 'optimal' case, allowing to (i) represent the set up as more ususally used (i.e. Not assimimation alsedo and P), and evaluated with the combination of an albedo and snow density parametrization agree the most with the observed SWE and SD.

We agree that this can be confusing in the manuscript. Therefore, to clarify this point:

- we have added the legend title "Input variables" to Fig. 3 to show which variables were used as input. These simulations resulted in the data with assimilated precipitation (Fig. 3e,f) and we have adjusted the caption.

- the term "optimal precipitation" has been replaced by "corrected precipitation" in Sect. 3.2.2 (line 168) to avoid confusion.

- a sentence at the end of Sect. 3.2.2 has been added to better explain this idealised case: *This idealised case corresponds therefore to simulations using the best possible combination of input data. As such observations are not always available or used to evaluate models, the idealised simulations are not used for the sensitivity study and model comparisons, which are based on optimal simulations (i.e. without assimilating precipitation and albedo, see Sect. 3.2.3). The simulated SWE and SD are compared to the observed SWE and SD and the assimilated precipitation data sets are shown. The precipitation data set ($P_{cor}$) that leads to a simulation with best correspondence to the observed SWE and SD is used in the further study.*

**Minor comments regarding Section 4.1**

> L237: If possible, please provide some details of the crashes (temperature related, model bugs?)

We did not investigate the crashes any further, but the crashes are likely related to out-of-bound temperatures. We have added to the text:
*The reason for these crashes has not been further investigated.*

> L249: It looks like that there was a high wind speed event mid-July. That certainly could have resulted in some erosion in reality, which is not simulated by the models. I think that that is as much an explanation as the overestimation of PSWE at the end of June.

We agree that a similar process occured, but also note that even if snow erosion was activated in SNOWPACK, this erosion was not simulated. We added in this section:

*Likewise, high wind speeds and a strong SWE and SD decrease is observed mid-July, which is likely snow erosion and also not considered in our simulations.*

> L251-252: As I mentioned earlier, it looks like melt is overestimated (since there is a lot of runoff simulated around this time when looking at Fig. 6c,d). So then the models need to simulate precipitation to bring the SWE/SD back to observational levels.

We agree with your comment. Indeed, melt is very strong in September. It is likely due to a low albedo not simulated by the model (see Fig. 4e and f). It might be attributed to a snow covered by dust transport by some strong wind events at the end of September, often observed in the field. The presence of dust, by darkening the snow surface, lead to a decrease in the albedo and an accelerating melt, which is not considered in these simulations. To compensate this effect, models try to readjust the precipitation amount, in other words, the models try to compensate processes not considered in this study. This is clarified in the manuscript:

*This is related to a strong melt in September, not simulated by the model, along with models trying to get the SWE and SD to the observational levels. The strong melt in September is caused by a sudden decrease of the albedo (observations in Fig. ??), as it is likely that the snow got covered with dust after some days with strong wind at the end of September, but the simulations overestimate the melt caused by this albedo decrease.*

> I struggle to understand the legend in Fig. 3, particularly when it comes to what the blue line represents, since it includes both "uncorrected precipitation" and SD/SWE, where I assume it's actually corrected precipitation by using the model's ability to assimilate SD/SWE?? I think at least the figure caption should be improved here. The figure caption should also explain why SnowModel has multiple blue and red lines.

The blue line shows the assimilated precipitation, if the observed precipitation and SD is given as input. Along with adding the title "Input variables" to the legends and changing "Uncorrected precipitation" into "Observed precipitation" in Fig. 3, we changed the caption of the Fig. 3:

*a-b) SD, c-d) SWE and e-f) the cumulative assimilated precipitation for the simulations with SNOWPACK (a,c,e) and SnowModel (b,d,f) and observations (black). The different input variables are given in the legenda. The dotted line of SWE indicates the less reliable (lower) SWE measurement from thallium rays (See Sect. 2.3) and the dotted line in e-f) is $P_{cor}$. The models have assimilated the observed precipitation (black) to the output (red/blue) given in e-f). Only one red and one blue line is shown for SNOWPACK as the other eight simulations crashed. The simulations for $z_0 = 1\,mm$ are found in Sect. S2.*

> I strongly recommend to add the undercatch correction from Wolff et al. (2015) in Panels 3e and 4f.

We think the reviewer meant to say Fig. 3e and 3f, as 4f is a figure with albedo, but we have added $P_{cor}$ to Fig. 3e and 3f and clarified this in the caption of the figure.

**Minor comments**

> L3/4: "While many studies focus on evaluating these uncertainties, issues still arise, especially in environments where sublimation is the main ablation process." "issues still arise" is very vague language, please improve this sentence with some concrete examples.

We clarified the text:

*While many studies focus on evaluating these uncertainties, no snow model comparison has been done in environments where sublimation is the main ablation process.*

> L60-65: Instead of only listing those studies, please provide details of what those studies found (particularly the results relevant to this study).

The studies in this section are only mentioned to show that an accurate assessment of different snow models' sensitivity to parameterization choice or input forcing is currently missing, although it is expected to have a large impact. Furthermore, some of the mentioned studies (Gascoin et al., 2013; Réveillet et al., 2020) are elaborated in the discussion.

> L90: unshielded

Done.

> L127: "includes *the* MeteoIO preprocessing library"

Done.

> L143: "new snow depth" This is somewhat confusing, since new snow depth can also be interpreted as the increase in depth from precipitation. Maybe "remaining snow depth"?

The text now reads:
*In SnowModel, the melted snow is redistributed over the remaining snow depth up to a maximum density threshold of 550 kg m$^{-3}$.*

> L157: "models were calibrated". The examples given indicate that it is not a calibration. If the soil albedo is set to 0.15 based on measured albedo, I would not consider that a calibration. I interpret calibration as trying different values and see what matches SWE or SD best.

We adjusted the text:
*Initially, both models were set up using similar parameters to facilitate intercomparison.*

> L256: overestimation of what exactly?

The overestimation of SWE:
*The overestimation of and the need for SWE as validation data are indications that the PSWE is not a valid precipitation dataset for our simulations, (..)*

> Fig 3: the x-tics in (a) do not align with the x-tics in the other subfigures. Please correct, since the figure suggests a common x-axis.

Done.

> L275: I actually struggle to see this in Fig. 4a,g.

We agree that it is hard to see the compaction in Fig. 4g. We think this is better visible in Fig. 4a, as the thicker red line in this figure has a flat slope, whereas the other runs show bigger increases during snowfall and faster decreases afterwards. We have added an explanation to help the reader:
*(i.e. the bold red line has a moderate slope, compared to the light red lines in Fig. ??a, until July)*

> L308: "e.q." -> "e.g."

Done.

> L328: Since for SnowModel, you talk about clusters, I think it's appropriate to talk about clusters here too. Instead of using the language of "range between 1.41 and 2.96", since it's not really a range. When I interpret it correcly, SNOWPACK produces two clusters in sublimation rate, based on roughness length.

Indeed, for SNOWPACK it was also clustered based on the $z_0$. The text has been adjusted:
*For SNOWPACK, the spread of the averaged sublimation rates corresponding to the ensemble runs from the first day of snow to 20 September have a minimum of 1.41 and a maximum of 2.96 mm w.e. d$^{-1}$ (Fig. 6a). During the cold period, when no melt occurs, the sublimation amounts mainly depend on the $z_0$, with sublimation rates ranging between 1.40 and 3.18 mm w.e. d$^{-1}$, but this is mainly clustered according to the implemented $z_0$. At the end of the season, the total sublimation ranges between 153 and 364 mm w.e. (corresponding to 36.2 to 86.0% of the total ablation, again stronly depending on the $z_0$).*

> L339: "Z0" -> "z0"

Done.

> L365: Since the main objective of the study seems to be to quantify sublimation, and sublimation is most strongly impacted by roughness length (see the dicussion on the clusters found in model results), I think that should be discussed before albedo.

The topics in the discussion have been reordered. The turbulent fluxes are discussed first, followed by the snow settling and albedo.

> Throughout manuscript: instead of talking about with and without precipitation uncertainties, I suggest writing including and excluding precipitation uncertainties. I think that makes it more clear.

This has been adjusted throughout the manuscript and also been changed in Fig. 5.

> L356: "such as grain size and snow surface area". It's actually called "specific surface area", but please note that SNOWPACK currently does not consider SSA specifically, since it's microstructure model is constructed based on totally different parameters. I suggest writing "grain size and microstructure".

The text now reads:
*Also, SNOWPACK considers a more complex representation of snow physics, such as the grain size and microstructure, (..)*

> L379: "as observations are biased". It's bit vague which observations are meant here. I assume authors mean that deriving density from SWE/SD measurements is biased over direct density observations using manual measurements?

It seems that this statement was too vague. We therefore considered you comment in clarifying this statement as follows:
*(..), but an improvement of snow density parameterizations in semi-arid regions shows a demand for snow density measurements, as deriving density from SWE/SD measurements is biased over direct density observations using manual measurements (Smith et al., 2017).*

> L495: Please update the SNOWPACK repository link. It seems to have changed recently.

Thank you for pointing out this. This has been updated in the revised version:
*SNOWPACK is an Open Source model and can be accessed at https://gitlabext.wsl.ch/snow-models/snowpack.*

**References**

Gascoin, S., Lhermitte, S., Kinnard, C., Bortels, K., and Liston, G. E.: Wind effects on snow cover in Pascua-Lama, Dry Andes of Chile, Advances in Water Resources, 55, 25–39, doi: 10.1016/j.advwatres.2012.11.013, 2013.

MacDonald, J. and Pomeroy, J.: Gauge undercatch of two common snowfall gauges in a prairie environment, in: Proceedings of the 64th Eastern Snow Conference, vol. 29, pp. 119–126, 2007.

Réveillet, M., MacDonell, S., Gascoin, S., Kinnard, C., Lhermitte, S., and Schaffer, N.: Impact of forcing on sublimation simulations for a high mountain catchment in the semiarid Andes, The Cryosphere, 14, 147–163, doi: 10.5194/tc-14-147-2020, 2020.

Rutter, N., Essery, R., Pomeroy, J., Altimir, N., Andreadis, K., Baker, I., Barr, A., Bartlett, P., Boone, A., Deng, H., Douville, H., Dutra, E., Elder, K., Ellis, C., Feng, X., Gelfan, A., Goodbody, A., Gusev, Y., Gustafsson, D., Hellström, R., Hirabayashi, Y., Hirota, T., Jonas, T., Koren, V., Kuragina, A., Lettenmaier, D., Li, W.-P., Luce, C., Martin, E., Nasonova, O., Pumpanen, J., Pyles, R. D., Samuelsson, P., Sandells, M., Schädler, G., Shmakin, A., Smirnova, T. G., Stähli, M., Stöckli, R., Strasser, U., Su, H., Suzuki, K., Takata, K., Tanaka, K., Thompson, E., Vesala, T., Viterbo, P., Wiltshire, A., Xia, K., Xue, Y., and Yamazaki, T.: Evaluation of forest snow processes models (SnowMIP2), Journal of Geophysical Research, 114, doi: 10.1029/2008jd011063, 2009.

Smith, C. D.: Correcting the wind bias in snowfall measurements made with a Geonor T-200B precipitation gauge and alter wind shield, in: 87th American Meteorological Society Annual Meeting, San Antonio, TX, 2007.

Smith, C. D., Kontu, A., Laffin, R., and Pomeroy, J. W.: An assessment of two automated snow water equivalent instruments during the WMO Solid Precipitation Intercomparison Experiment, The Cryosphere, 11, 101–116, doi: 10.5194/tc-11-101-2017, 2017.

Wolff, M. A., Isaksen, K., Petersen-Øverleir, A., Ødemark, K., Reitan, T., and Brækkan, R.: Derivation of a new continuous adjustment function for correcting wind-induced loss of solid precipitation: results of a Norwegian field study, Hydrology and Earth System Sciences, 19, 951–967, doi: 10.5194/hess-19-951-2015, 2015.

---

## Author Response (AR3)

**Response to review #3**
Snow model comparison to simulate snow depth evolution and sublimation at point scale in the semi-arid Andes of Chile

Annelies Voordendag, Marion Réveillet,
Shelley MacDonell, Stef Lhermitte

August 11, 2021

Dear editor,

First, we would like to thank the editor for his careful third evaluation of our work and the detailed suggestions and comments. Below we adress our detailed responses to all the comments.

As for the first rounds of reviews, in this response-to-review document we clarified and addressed each of the editor's suggestions and comments. Therefore we have copied the comments in blue boxes and have addressed them one by one. In the response we use italic fonts to quote text from the revised manuscript. Additional to the revised manuscript, we have uploaded a supplementary version of the manuscript with highlighted track changes that indicate where the manuscript has changed (red=removed; blue=added).

Yours sincerely, Annelies Voordendag & co-authors

**Response to the Editor B. Noel**

> Dear Annelies Voordendag and co-authors,
>
> Thank you for submitting your revised manuscript to TC. The authors provided convincing answers to reviewer #3. The manuscript could still benefit from a few clarifications, and the editor recommends publication in TC after applying/considering the minor comments/suggestions below. Note that the line numbering is based on the "tracked changes" version of the revised manuscript.
>
> The editor will re-assess the revised manuscript before acceptance in TC. Best wishes,
>
> Brice Noël

The authors thank the editor for his new careful reading and his feedback. As requested, in this revised version of the manuscript, along with accounting for the suggestions, we briefly present the main findings and limitations of the studies performed by (Gascoin et al., 2013) and (Réveillet et al., 2020) in the introduction. We also clarified the meaning of $P_{cor}$.

**Editor's minor comments**

> 1. The editor agrees with reviewer #3 that the authors should briefly present the main findings of (1) Gascoin et al. (2013) and (2) Réveillet et al. (2020) in the introduction. These two studies present (1) a process that is not accounted for in this study (i.e. snow drift) and (2) another estimate of the relative contribution of melt/sublimation to total ablation in the area.

More details about these studies have been added in the reviewed version, following your comment. You can now read:

*In previous studies, Gascoin et al. (2013) assessed the effect of wind transport on snow cover in the semi-arid Andes using numerical simulations with SnowModel (Liston and Elder, 2006), and highlighted the significant importance of blowing snow sublimation. They also evidenced the difficulty of the model to capture the small-scale snow depth spatial variability, partly related to the lack of reliable input data such as precipitation. Réveillet et al. (2020) indicated that ablation is dominated by sublimation in the semi-arid Andes and that the sublimation ratio increases with elevation. They also quantified a similar proportion of sublimation ratio for two years with contrasting climatic*

*conditions (i.e. dry versus wet), but pointed out the significant uncertainties related to the forcing. The study performed by Mengual Henríquez (2017) assessed the snow types in different Chilean regions with SNOWPACK (Bartelt and Lehning, 2002; Lehning et al., 2002a,b) and mainly found that SNOWPACK is a powerful snow model, but an improvement of the forcing data is needed to improve simulations.*

> 2. In L189-192: The authors should clarify that the precipitation correction of "Pcor" used in "optimal" simulations is based on Wolff et al. (2015).

The text now reads:
*Due to complexities with the assimilated precipitation data and the need for SWE as validation data, the precipitation data set ($P_{cor}$) that is used in the further study is based on a wind correction by Wolff et al. (2015) (See Sect. 4.1).*

> 3. For clarity, the authors should verify that "P" is used for observed (uncorrected) precipitation and "Pcor" for the optimal (corrected) precipitation data used in the sensitivity experiments (including both parameterization and perturbed forcing). For instance, "P data" in L210 and "P" in L217 should better be replaced by "Pcor". In addition, the authors could add "(P)" after "Precipitation forcing" in L89.

The proposed adjustments have been made, along with adding $P_{cor}$ in the caption of Table 3.

> 4. The authors should clearly mention in the caption of all Figures showing SWE that the black solid (dotted) line represents measurements from potassium (thallium).

We have done this for Figures 3, 4, 5, S2.1 and S3.1.

**Editor's specific comments**

> L101: For clarity, the authors could replace "accuracy" by "uncertainty".

Done.

> L177: This sentence remains unclear. Do the authors mean "This cumulative SWE approach does not account for snow drift deposition, which inclusion would have resulted in ... "?

The text now reads:
*The positive SWE changes beyond precipitation events are not accounted for, as they might originate from deposition caused by snow drift and its inclusion would have resulted in an overestimation in this data set.*

> Figure 2 caption: "air temperature (TA)".

Done.

> Figure 3 caption: "The solid (dotted) line in c-d) indicates more (less) reliable SWE measurement from potassium (thallium) rays ...". See also minor comment #4.

Done. The caption now reads: *a-b) SD, c-d) SWE and e-f) the cumulative assimilated precipitation for the simulations with SNOWPACK (a,c,e) and SnowModel (b,d,f) and observations (black). The different input variables are given in the legenda. The solid (dotted) line in c-d) indicates the more (less) reliable SWE measurement from potassium (thallium) rays (See Sect. 2.3) and the dotted line in e-f) is $P_{cor}$. The models have assimilated the observed precipitation (black) to the output (red/blue) given in e-f). Only one red and one blue line is shown for SNOWPACK as the other eight simulations crashed. The simulations for $z_0 = 1\,mm$ are found in Sect. S2.*

> Figures 4-5 caption: See previous comment and minor comment #4.

Done.

> Figure S2.1 caption: "red line is visible in e-f)."; "the dotted line in c-d) ...".

The caption now reads:

*a-b) SD, c-d) SWE and e-f) the cumulative assimilated precipitation for SNOWPACK (a,c,e) and SnowModel (b,d,f) and observations (black). The different forcing parameters are given in the legenda. The simulations with SNOWPACK for every different input set were done with five different fresh snow density parameters and the simulations with SnowModel for every input set were done with six combinations out of three fresh snow density and two albedo parameterizations. PSWE is equal for $z_0$ is 1 mm and 1 cm and thus only the red line is visible in e-f). The solid (dotted) line in c-d) indicates the more (less) reliable SWE measurement from potassium (thallium) rays.*

Figure S3 caption: "precipitation from SWE (PSWE) ..."

We have added precipitation constructed from SWE:
*Observed cumulative precipitation, precipitation corrected from SWE (PSWE) and precipitation corrections (MacDonald and Pomeroy, 2007; Smith, 2007; Wolff et al., 2015). The two SWE observations with potassium (K) and thallium (Tl) gamma rays are also given.*

**Editor's suggestions**

L65: "Despite these previous studies," instead of "Nevertheless,".

Done.

L99: The editor suggests: "but differences of up to ... between potassium and thallium gamma ray measurements at 300 ...".

Done.

L150: For consistency with L157, replace "calibrated" by "set up".

Done.

L159: "observed surface albedo when there is no snow cover.".

Done.

L163: Replace "first snow" by "fresh snow".

Done.

L168: "Therefore, to correct the precipitation used as input for the models, ..."

Done.

L190: Add "in Fig. 3" after "precipitation data sets are shown".

Done.

L224-225: "and the difference between the ... cover approximately 200 mm w.e. (Sect 4.1)."

Done.

L250: "align".

Done.

L262: "in Fig. 4e-f), as it is ...".

Done.

L264: "decrease are observed ... which is likely due to snow erosion that is not considered ...".

Done.

> L288: "fresh snow density parameterization".

Done.

> L320: "shows a similar".

Done.

> L339: The editor suggests "Impacts" instead of "Consequences".

Done.

> L347: "strongly".

Done.

> L419: The editor suggests "evaluated" instead of "verified".

Done.

**References**

Bartelt, P. and Lehning, M.: A physical SNOWPACK model for the Swiss avalanche warning: Part I: Numerical model, Cold Regions Science and Technology, 35, 123–145, doi: 10.1016/s0165-232x(02)00074-5, 2002.

Gascoin, S., Lhermitte, S., Kinnard, C., Bortels, K., and Liston, G. E.: Wind effects on snow cover in Pascua-Lama, Dry Andes of Chile, Advances in Water Resources, 55, 25–39, doi: 10.1016/j.advwatres.2012.11.013, 2013.

Lehning, M., Bartelt, P., Brown, B., and Fierz, C.: A physical SNOWPACK model for the Swiss avalanche warning: Part III: Meteorological forcing, thin layer formation and evaluation, Cold Regions Science and Technology, 35, 169–184, doi: 10.1016/s0165-232x(02)00072-1, 2002a.

Lehning, M., Bartelt, P., Brown, B., Fierz, C., and Satyawali, P.: A physical SNOWPACK model for the Swiss avalanche warning: Part II: Snow microstructure, Cold Regions Science and Technology, 35, 147–167, doi: 10.1016/s0165-232x(02)00073-3, 2002b.

Liston, G. E. and Elder, K.: A Distributed Snow-Evolution Modeling System (SnowModel), Journal of Hydrometeorology, 7, 1259–1276, doi: 10.1175/jhm548.1, 2006.

MacDonald, J. and Pomeroy, J.: Gauge undercatch of two common snowfall gauges in a prairie environment, in: Proceedings of the 64th Eastern Snow Conference, vol. 29, pp. 119–126, 2007.

Mengual Henríquez, S. A.: Caracterización de la nieve de distintas localidades de Chile mediante el uso del modelo SNOWPACK, Master's thesis, Universidad de Chile, 2017.

Réveillet, M., MacDonell, S., Gascoin, S., Kinnard, C., Lhermitte, S., and Schaffer, N.: Impact of forcing on sublimation simulations for a high mountain catchment in the semiarid Andes, The Cryosphere, 14, 147–163, doi: 10.5194/tc-14-147-2020, 2020.

Smith, C. D.: Correcting the wind bias in snowfall measurements made with a Geonor T-200B precipitation gauge and alter wind shield, in: 87th American Meteorological Society Annual Meeting, San Antonio, TX, 2007.

Wolff, M. A., Isaksen, K., Petersen-Øverleir, A., Ødemark, K., Reitan, T., and Brækkan, R.: Derivation of a new continuous adjustment function for correcting wind-induced loss of solid precipitation: results of a Norwegian field study, Hydrology and Earth System Sciences, 19, 951–967, doi: 10.5194/hess-19-951-2015, 2015.